# Highly supercooled riming and unusual triple-frequency radar signatures over McMurdo station, Antarctica

Frederic Tridon[1,*], Israel Silber[2], Alessandro Battaglia[3,4,5], Stefan Kneifel[1,6], Ann Fridlind[7], Petros Kalogeras[4,5], and Ranvir Dhillon[4]

[1]Institute for Geophysics and Meteorology, University of Cologne, Cologne, Germany
[2]Department of Meteorology and Atmospheric Science, Pennsylvania State University, University Park, PA, USA
[3]DIATI, Politecnico di Torino, Turin, Italy
[4]University of Leicester, Leicester, UK
[5]National Center for Earth Observation, Leicester, UK
[6]Meteorological Institute, Ludwig-Maximilians-University Munich, Munich, Germany
[7]NASA Goddard Institute for Space Studies, New York, NY, USA
[*]Now at: DIATI, Politecnico di Torino, Turin, Italy

**Correspondence:** frederic.tridon@polito.it

**Abstract.** Riming of ice crystals by supercooled water droplets is an efficient ice growth process, but its basic properties are still poorly known. While it has been shown to contribute significantly to surface precipitation at mid-latitudes, little is known about its occurrence at high latitudes. In Antarctica, two competing effects can influence the occurrence of riming: the scarcity of supercooled liquid water clouds due to the extremely low tropospheric temperatures and the low aerosol concentration,

which may lead to the formation of fewer and larger supercooled drops potentially resulting in an enhanced riming efficiency.

In this work, by exploiting the deployment of an unprecedented number of multi-wavelength remote sensing systems (including triple-frequency radar measurements) in West Antarctica, during the Atmospheric Radiation Measurements West Antarctic Radiation Experiment (AWARE) field campaign, we evaluate the riming incidence at McMurdo station and find that riming occurs at much lower temperatures compared to previous results in the mid-latitudes. This suggests the possibility of occur-

rence of a common atmospheric state over Antarctica that includes a rather stable atmosphere inhibiting turbulent mixing, and a high riming efficiency driven by large cloud droplets.

We then focus on a peculiar case study featuring a persistent layer with a particularly pronounced riming signature in triple-frequency radar data but only a relatively modest amount of supercooled liquid water. In-depth analysis of the radar observations suggests that such signatures can only be explained by the combined effects of moderately rimed aggregates or

similarly shaped florid polycrystals and a narrow particle size distribution (PSD). Simulations of this case study performed with a 1D bin model indicate that similar triple frequency radar observations can be reproduced when narrow PSDs are simulated. Such narrow PSDs can in turn be explained by two key factors: (i) the presence of a shallow homogeneous droplet or humidified aerosol freezing layer aloft seeding an underlying supercooled liquid layer, and (ii) the absence of turbulent mixing throughout a stable polar atmosphere that sustains narrow PSDs, as hydrometeors grow from the nucleation region aloft to several millimeter

ice particles, by vapor deposition and then riming.

# 1 Introduction

Besides deposition and aggregation, riming is an efficient ice growth process. It contributes significantly to surface precipitation at mid-latitudes (Grazioli et al., 2015; Moisseev et al., 2017) and is pivotal for improving our understanding of the role of ice phase in the water budget. However, basic properties of riming such as its efficiency or its importance in precipitation formation are still largely unknown since it involves the collection of poorly characterized supercooled water droplets by complex ice particles. It is widely accepted, however, that at the beginning of a riming process, the mass of a rimed ice particle increases while its maximum dimension remains constant or only slightly increases (e.g., Heymsfield, 1982; Seifert et al., 2019); hence, the density and fall speed of rimed ice hydrometeors tend to be enhanced. Riming occurrence is strongly linked with temperature since the probability of finding supercooled liquid water decreases with temperature. By exploiting a multi-year dataset of cloud radar observations at four European sites in various environment, Kneifel and Moisseev (2020) showed that riming is rare below -12°C and more frequent closer to 0°C.

In the Arctic, supercooled liquid water clouds are frequent (e.g., Shupe et al., 2008; Cesana et al., 2012; Morrison et al., 2012; Mioche et al., 2015) and rimed precipitating particles are commonly observed (Mioche et al., 2017; Fitch and Garrett, 2022). In Antarctica however, liquid water clouds are less frequent, in particular during winter months due to lower temperatures (e.g., Matus and L'Ecuyer, 2017; Lubin et al., 2020). Nevertheless, the typical low aerosol concentrations in this region can lead to the formation and persistence of supercooled drizzle drops (Silber et al., 2019a), which might facilitate the occurrence of riming due to the enhanced riming efficiency of drizzle drops (Lohmann, 2004). Therefore, a thorough investigation of riming in Antarctic clouds is timely.

Field measurements in Antarctica are historically sparse due to logistical challenges. Space-borne instruments such as the Cloud Profiling Radar (CPR) onboard CloudSat (Stephens et al., 2008) can cover extended and remote areas but have inherent limitations to measure weak ice precipitation fluxes (e.g., Silber et al., 2021), or any ice precipitation fluxes near the ground due to the so-called "blind-zone" (Maahn et al., 2014). Only recently, riming has been shown to be a recurring process at an Antarctic site based on ground-based optical probe observations at the Dumont d'Urville Station (Grazioli et al., 2017a), with most of the detected large ice hydrometeors being at least partially rimed. However, in order to detect an active riming process, suitable measurement are needed across the vertical column, which can be achieved via ground-based multi-frequency radars, for example.

By analysing scattering models of snow aggregates and graupel, Kneifel et al. (2011) suggested that triple-frequency radar measurements could be exploited to differentiate between rimed and unrimed ice particles. This differentiation has been later verified by comparing triple-frequency radar signatures with bulk snow density derived from collocated ground-based observations (Kneifel et al., 2015). While the radar Doppler velocity is the simplest and most obvious parameter for retrieving the degree of riming or an equivalent parameter (e.g. density factor or rime mass fraction) of ice particles (e.g., Mosimann, 1995; Mason et al., 2018; Kneifel and Moisseev, 2020), triple frequency radar observations can also provide critical information on the internal structure of snowflakes (Mason et al., 2019), and hence, on the growth processes involved. For example, by exploiting triple-frequency Doppler spectra, Kneifel et al. (2016) combined triple-frequency and Doppler velocity information and

confirmed that rimed and unrimed aggregates produce distinct scattering signatures. Further development of multi-frequency radar retrievals demonstrated that the combination of three radar frequencies enables the derivation of snow aggregate properties with various degree of riming (e.g., Mason et al., 2018). Quantitative agreement was found with the measurements from collocated ground-based (Moisseev et al., 2017; von Lerber et al., 2017) or airborne (Leinonen et al., 2018; Tridon et al., 2019) in-situ probes.

In the framework of the Atmospheric Radiation Measurement (ARM) West Antarctic Radiation Experiment (AWARE, Lubin et al., 2020), the U.S. Department of Energy (DOE) deployed the second ARM Mobile Facility (AMF2) at McMurdo Station from 1 December 2015 to 31 December 2016, resulting in an unprecedented suite of remote sensing instruments in Antarctica, including the Ka-band ARM Zenith Radar (KAZR), the Marine W-band ARM Cloud Radar (MWACR) and the scanning dual-wavelength ARM cloud radar system (X/KaSACR). Although the MWCAR stopped transmitting after about three months, these instruments provided triple-wavelength radar profiles for the first time in Antarctica (in Sect. 2). In this work these data have been exploited to evaluate the probability of finding triple-frequency signatures of riming in clouds over McMurdo Station and have been compared with climatologies collected at other triple-frequency radar sites at mid-latitudes and in the Arctic (Sect. 3). A case study with strong triple-frequency signatures is further analysed in Sect. 4 via a detailed retrieval of ice microphysics, and bin model simulations performed to investigate its salient features. Conclusions are drawn in Sect. 5.

## 2 Radar observations during AWARE

### 2.1 The AWARE field campaign

The AWARE field campaign aimed to acquire critical atmospheric data to fundamentally understand atmospheric forcing on West Antarctica, and to foster related improvements to climate model performance (Lubin et al., 2020). It hinged upon the deployment of the AMF2 to McMurdo Station on Ross Island (77°50′47″S, 166°40′06″E, 76 m above mean sea level; see Fig. 2a) with the goal of sampling an annual cycle in atmospheric structure and thermodynamics, surface radiation budget and cloud properties. The AMF2 includes cloud research radars, lidars, multiple broadband and spectral radiometers, an aerosol observation suite, and thorough meteorological sampling instruments ranging from surface turbulent flux equipment to radiosondes. The present study is focused on the processing and interpretation of radar data.

The lack of orographic features in the Southern Ocean surrounding Antarctica supports the midlatitude westerlies, generally isolating the Antarctic region from moisture and aerosol sources (Lubin et al., 2020). Nevertheless, synoptic-scale low-pressure systems over the Ross Sea periodically inject marine air poleward over West Antarctica (Nicolas and Bromwich, 2011), and act as the main source of heat and moisture to the Ross Island region, impacting meteorological conditions at McMurdo (Silber et al., 2019b; Scott and Lubin, 2014). As such, the observations made during AWARE might be representative of various Antarctic coastal regions. On the contrary, the steep coastal slopes and high terrain of the East Antarctic Ice Sheet present a barrier to the penetration of marine air masses, where large-scale subsidence and low temperatures limit precipitable water vapor amounts resulting in lower cloud occurrences than is typical of maritime regions (Silber et al., 2018a).

During AWARE, temperatures and cloud fractions were comparable to long-term measurements reported by Monaghan et al. (2005) at McMurdo. Statistics during AWARE have been described in details in Silber et al. (2018a). The monthly mean temperatures between 0 and 4 km varied from -30°C in winter to -15°C in summer. The annual mean cloud fractions was 67%, significantly higher than at the South Pole Station. More details about the environmental characteristics at McMurdo during AWARE can be found in Silber et al. (2018a), Zhang et al. (2019) and Lubin et al. (2020).

## 2.2 Radar data processing

In this study, we exploit the data collected by KAZR, MAWCR, radiosonde and XSACR data (Atmospheric Radiation Measurement (ARM) user facility, 2014, 2015a, b, c, respectively). While the KAZR and MWACR are zenith-pointing radars, the X/KaSACR loops through a sequence of various scanning modes in order to sample the three-dimensional geometry of clouds (Kollias et al., 2014), including a zenith-pointing period of about 25 min every 2 hours. Triple-frequency radar observations are therefore available only during these zenith-pointing operation periods. In this study, KAZR data was prefered to KaSACR data because of its better sensitivity. At the beginning of the field campaign, the radar beams alignment has been maximized for an optimal volume matching. Since the temporal and range resolution of the radars slightly differ, their data have been first regridded to a common 3 s by 30 m time–height grid.

Following standard ARM procedures, absolute calibrations of the scanning radar systems have been performed on site with a corner reflector and the calibration of the KaSACR has been transferred to the KAZR via a statistical comparison of the reflectivities measured in the vertical (Kollias et al., 2016, 2020). Without the possibility to use natural volume targets, such as rainfall, for checking the radar calibration (e.g. involving a co-located disdrometer as in Dias Neto et al., 2019), the calibration cannot be considered to be more accurate than $\pm 3$ dB and absolute reflectivities are mainly used qualitatively in the current study. The KAZR calibration provided in the ARM Archive was deemed appropriate despite the results from Kollias et al. (2019), based on a systematic comparison with nearby measurements from Cloudsat, suggesting a rather large miscalibration of the KAZR during AWARE. Indeed, such an automatic method is challenging in an area with complex topography like McMurdo and, for the AWARE campaign, it suggests an erratic KAZR calibration instability with an offset ranging between 3.5 and 7.7 dB. Furthermore, thanks to coincidental observations during the case study presented in Sect. 4, comparisons of KAZR and Cloudsat reflectivities suggest the ARM calibration to be appropriate.

Before deriving the dual-wavelength ratios (DWRs), the relative calibration between the different radars is performed. Firstly, the two-way attenuation profile due to atmospheric gases is derived from the measurements of the closest radio soundings and the absorption model of Rosenkranz (1998). Secondly, the remaining offsets due to supercooled liquid, snow, and radome attenuation as well as possible absolute calibration differences are derived by matching the measured reflectivity near cloud tops, where only small hydrometeors are present and non-Rayleigh scattering is negligible (Tridon et al., 2020). While the XSACR calibration proposed in the ARM Archive was found to be correct, a considerable offset of +19.6 dB was necessary for the MWACR.After calibration, the sensitivity at 1 km level and 2 s integration time are -40, -52 and -36 dBZ for the XSACR, KAZR and MAWCR, respectively.

**Table 1.** List of AWARE cases with triple frequency radar observations and corresponding mean environmental variables. Temperature and wind are averages over the 0 to 5 km AGL layer obtained from radiosondes measurements. Relative humidity is measured with an ARM surface meteorological station.

| Start time [UTC] | End time [UTC] | Duration with 3-frequency [min] | Cloud top [km AGL] | Temperature [°C] | RH [%] | Wind [m s$^{-1}$] |
|---|---|---|---|---|---|---|
| 31/12/2015 17:00 | 01/01/2016 21:00 | 46 | 3.5 | -22.2 | 79 | 5.7 |
| 02/01/2016 10:00 | 05/01/2016 09:00 | 351 | 6 | -21.0 | 80 | 9.8 |
| 09/01/2016 23:00 | 11/01/2016 15:00 | 117 | 8 | -17.6 | 83 | 13.7 |
| 16/01/2016 13:00 | 21/01/2016 07:00 | 339 | 7 | -15.4 | 82 | 8.1 |
| 28/01/2016 14:00 | 01/02/2016 17:00 | 172 | 6 | -20.2 | 60 | 11.9 |
| 02/02/2016 18:00 | 03/02/2016 08:00 | 86 | 6 | -19.1 | 59 | 16.4 |
| 08/02/2016 18:00 | 12/02/2016 09:00 | 180 | 5 | -23.0 | 58 | 6.5 |

Due to a failed power supply, the MWACR was taken offline in March 2016 (Lubin et al., 2017) and the triple-frequency dataset is limited to only about three months. Nevertheless, during this period 7 multi-day snowfall events were recorded during which the signal to noise ratio of all three radars exceeded -10 dB. This results in a total duration of 21 hours of triple frequency observations (see Table 1), providing insights on how frequent riming might be in Antarctica, at least for the summer season.

## 3 Triple-frequency signatures during AWARE

### 3.1 Results from previous data sets

In order to highlight the occurrence of aggregation or riming processes, it is helpful to combine the DWRs of all three frequencies in a single plot showing DWR$_{X,Ka}$ as function of DWR$_{Ka,W}$, as proposed by Kneifel et al. (2011). When snowflakes become sufficiently large (with a threshold on the characteristic sizes that depend on the frequency pair; see Fig. A4 in Battaglia et al., 2020a), their reflectivity depends on the radar frequency and the DWRs depart from zero. In a nutshell, the DWR$_{X,Ka}$ and DWR$_{Ka,W}$ increase almost equally in case of aggregates, while the DWR$_{X,Ka}$ remains much lower than DWR$_{Ka,W}$ in case of rimed particles (a maximum DWR$_{X,Ka}$ of roughly 3 dB was suggested by Kneifel et al., 2015; Dias Neto et al., 2019, but it can reach slightly larger values when the mean mass diameter is larger than 3 mm). The proposed explanation for this behaviour is that the rimed particles are too small to enhance the DWR$_{X,Ka}$ while their larger density enhance their refractive index, and hence, the DWR$_{Ka,W}$ (Dias Neto et al., 2019). In case of very large low-density aggregates, the DWR$_{Ka,W}$ can actually decrease producing a bending back of the curve (for details, see Kneifel et al., 2015). Mason et al. (2019) have shown that the shape of the size distribution and the internal structure of snowflakes also have a non-negligible influence on the triple-frequency signatures.

The ARM program has pioneered ground-based triple-frequency radar observations (e.g., during the BAECC field campaign, Petäjä et al., 2016) and similar experimental setups are emerging at other sites such as the TRIple-frequency and Polarimetric radar experiment (TRIPEx) at the Jülich Observatory for Cloud Evolution, Germany (Dias Neto et al., 2019). The triple-frequency density occurrence derived from datasets collected at these sites generally include the branches of both aggregates and rimed particles (Kneifel et al., 2015; Mason et al., 2018, 2019; Dias Neto et al., 2019). This is also true for the AWARE snowfall event on the $10^{th}$ of January which was succinctly analysed in Lubin et al. (2020). A peculiarity of the AWARE case analysed in the current paper (see Sect. 4) is the presence of a rimed particle branch leading to very large $DWR_{Ka,W}$ values (up to 16 dB); $DWR_{Ka,W}$ barely exceeds 12 dB for all other studies cited above and corresponding to data from various sites located at mid to high latitudes.

## 3.2   Temperature dependence of DWRs

In order to investigate the conditions at which the aggregation and riming processes occur, another way of showing the triple-frequency signatures is to plot the profiles of the observed DWRs after they have been stratified according to air temperature (Fig. 1a,b), as suggested by Dias Neto et al. (2019) for their TRIPEx dataset. To this aim, the temperature information has been interpolated from the closest radio soundings, which were launched every 12 hours during AWARE.

For comparison, the same methodology (Fig. 1c,d) has also been applied to the triple-frequency dataset of the BAECC field campaign (Petäjä et al., 2016), during which the ARM program deployed the AMF2 (i.e., the same instruments as for AWARE) at the Hyytiälä Field Station of the University of Helsinki, Finland (61°50'37.114"N, 24°17'15.709"E, 150 m above mean sea level) from 1 February to 12 September 2014. During the winter season, 20 snowfall events were recorded, resulting in 35 hours of triple frequency radar data. Most of the snowfall was associated to deep frontal systems bringing moist air from the Baltic Sea. The most important case studies have already been thoroughly analysed in previous papers (e.g., Kneifel et al., 2015; Kalesse et al., 2016; von Lerber et al., 2017; Moisseev et al., 2017; Mason et al., 2018, 2019; Tridon et al., 2020). Interestingly, the BAECC DWR density plots are practically identical to those from TRIPEx (Fig. 9 in Dias Neto et al., 2019), both sites representing well the northern hemisphere mid-latitudes.

Despite appearing a bit noisier due to the slightly reduced size of the dataset (i.e. 21 vs. 35 hours), the AWARE density plots (Fig. 1) show interesting similarities in comparison with those from BAECC (and, equivalently, TRIPEx), but also some striking differences. On the similarities side, the medians of both DWRs (black lines) reach nearly the same maxima around 0°C (6 and 2 dB for $DWR_{Ka,W}$ and $DWR_{X,Ka}$, respectively). Furthermore, the rate of increase of the $DWR_{X,Ka}$ with temperature is similar: it remains small at low temperatures and increases faster for temperatures greater than -15°C, which can be explained by a rapid growth of aggregates favored by the dendritic growth around -15°C. On the disparities side, the AWARE $DWR_{Ka,W}$ increases at a lower temperature (around -25°C) compared to the mid-latitude sites. Both the median and the width of the distribution of $DWR_{Ka,W}$ increase significantly more for AWARE than for BAECC. Specifically between -25°C and -15°C, the median and width increase by 2.6 and 2.4 dB for AWARE, while they only increase by 1.2 and 1.8 dB for BAECC. This is a very peculiar feature which was never observed in the previous yet longer field campaigns during which triple-frequency radar measurements were collected. Furthermore, there is a secondary but striking branch of $DWR_{Ka,W}$ reaching extreme values

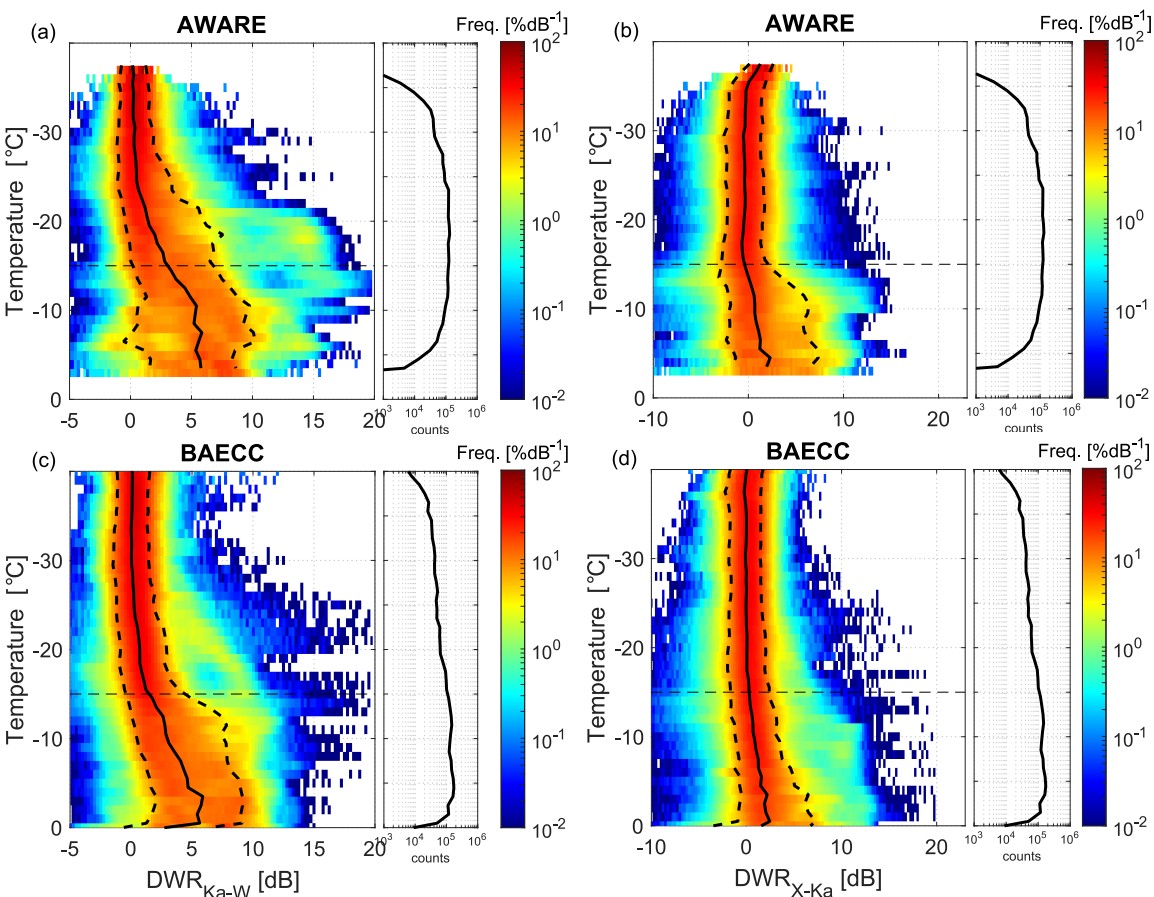

**Figure 1.** Density plots of $DWR_{Ka-W}$ (a,c) and $DWR_{X-Ka}$ (b,d) as function of temperature for AWARE (a,b) and BAECC (c,d) datasets. The dashed lines indicate the $10^{th}$ and $90^{th}$ percentiles. The subplots on the right of each panel indicate the number of samples per temperature level.

of 15 dB, much higher than the common maximum of 12 dB. Even if it only represents 6% of the AWARE triple frequency dataset, this corresponds to 75 min. It was observed during two different cases: the $2^{nd}$ to $4^{th}$ January 2016 case, which will be partly discussed in Sect. 4, and the $9^{th}$ to $10^{th}$ January 2016 case, which has already been described in Lubin et al. (2020).

If Ka and W-band were the only radar pair available, it could be argued that an enhancement of aggregation due to local dynamic effects could be the most probable process leading to the increase of $DWR_{Ka,W}$ at low temperature. However, the corresponding $DWR_{X,Ka}$ remains close to 0 dB, which can only be explained by the presence of rimed particles at lower temperatures during AWARE. While we cannot exclude a potential influence from vertical winds induced by the complex topography around McMurdo station, these differences from previous studies can more likely be explained by the low concentration of aerosol in Antarctica, compared to the northern hemisphere: a low cloud condensation nuclei concentration could

lead to fewer but larger supercooled droplets (for a given cloud water content), and therefore, more efficient riming (Lohmann, 2004). Indeed, even with a fewer number of droplets, the riming process can be favored in a clean environment because the collision efficiency between an ice crystal and a liquid droplet strongly increases when going from small cloud droplets to slightly larger drizzle (Pruppacher and Klett, 1996; Wang and Ji, 2000). In the rest of the paper, the occurrence of riming at low temperatures will be further assessed by focusing on the $2^{nd}$ to $4^{th}$ January 2016 case, which features the strongest $\text{DWR}_{Ka,W}$ of the AWARE dataset.

## 4 Extreme triple-frequency signatures of the $4^{th}$ January 2016

### 4.1 General description of the case study

Between the $1^{st}$ and $4^{th}$ January 2016, the weather conditions were typical of the frequent strong katabatic wind events recorded at McMurdo Station (Chenoli et al., 2013; Coggins et al., 2014; Monaghan et al., 2005; Weber et al., 2016). The Ross sea semi-permanent cyclonic circulation (Carrasco and Bromwich, 1994; Monaghan et al., 2005; Simmonds et al., 2003) deepened and moved to the South, bringing moist air over the Ross Ice Shelf (see Fig. 2a). MODIS cloud phase retrieval (see Fig. 3) indicates that clouds formed over the ice shelf, including extended clouds with supercooled liquid at their top. The evolution of cloud features in the subsequent panels of Fig. 3 demonstrates the cyclonic (clockwise) circulation centered around the North of the Ross ice shelf, which led to deeper ice-topped clouds along the Transantarctic Mountains to the West of the Ross ice shelf (see Fig. 2c,d). Cloud initiation mechanisms included lifting of air due to the relief barrier and convergence of cyclone winds with katabatic winds descending from the Antarctic Plateau. Between the $2^{nd}$ and $4^{th}$ of January, this resulted in strong Southerly winds (e.g. winds up to $16\,\text{m}\,\text{s}^{-1}$ were recorded at $1.7\,\text{km}$ ASL by the radiosonde launched from the AMF2 at McMurdo on the $4^{th}$ of January at 11h UTC as shown in Fig. 2b) associated with long lasting clouds deepening on the windward side of Ross Island.

On the $4^{th}$ January 2016, Cloudsat made two overpasses exceptionally close to McMurdo Station (as close as 46 and $23\,\text{km}$ at 5:16 and 11:47 UTC, respectively). Cloudsat reflectivity transects (Figs. 2c and d) confirm the presence of extended and complex cloud fields over the whole Ross Ice Shelf, and particularly deep clouds near McMurdo with cloud tops reaching nearly $6\,\text{km}$ ASL and reflectivity as large as $14\,\text{dBZ}$. Furthermore, co-located CALIPSO observations indicate that most of these clouds were mixed-phase clouds with a supercooled liquid layer at their tops at temperatures as low as -35°C (magenta lines in Figs. 2c and d).

### 4.2 Observations at McMurdo

Over McMurdo, a persistent thick cloud layer was continuously observed between the $2^{nd}$ and $5^{th}$ January 2016. While the ARM lidars were not able to penetrate through the full extent of the clouds and sample their top, the associated liquid cloud base height products (Silber et al., 2018b, c) suggest that supercooled liquid layers were almost always present at various heights within the clouds, from 0.5 to $3\,\text{km}$ AGL for the whole three days (not shown).

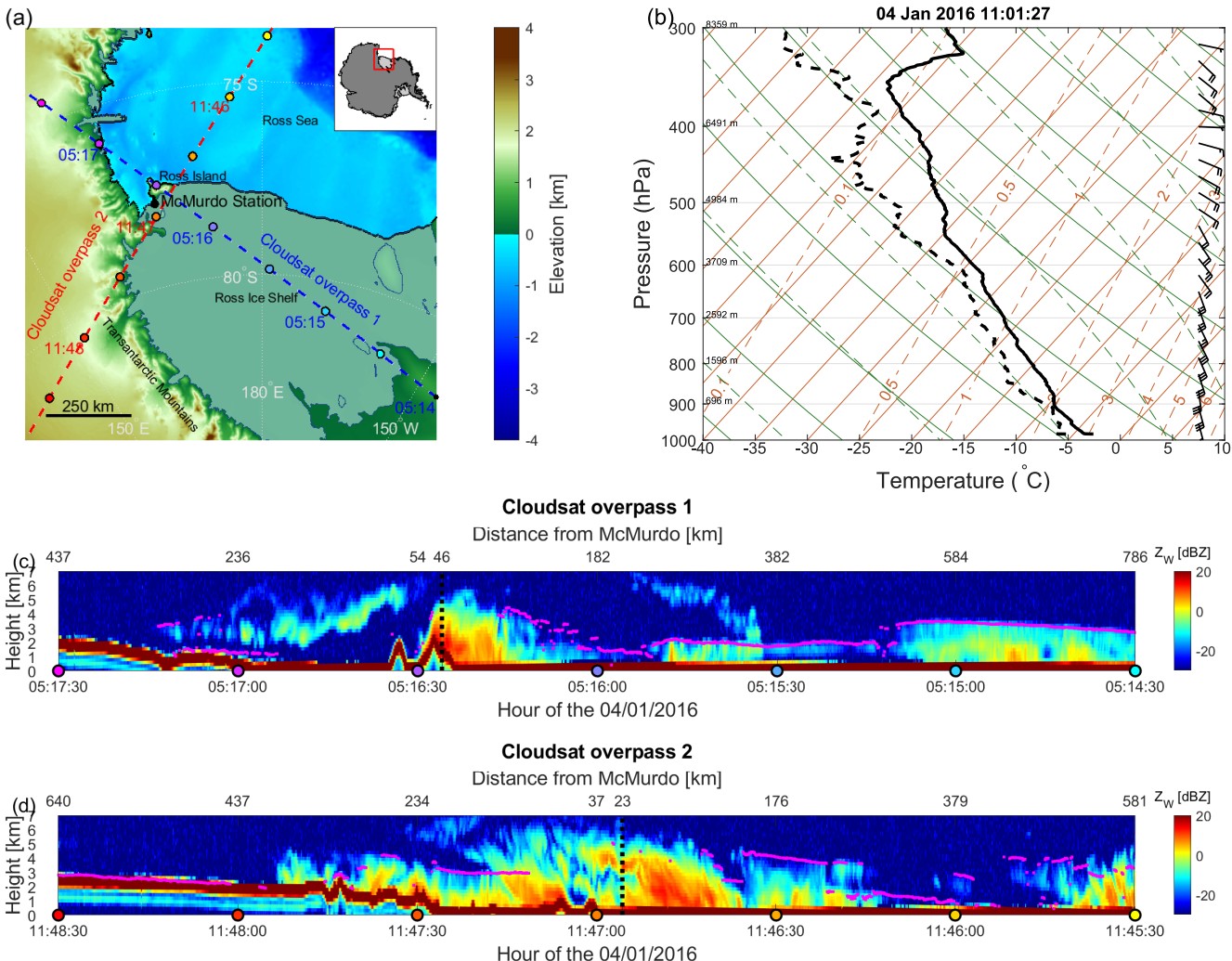

**Figure 2.** (a) Overview of the geographical features around McMurdo created with the Antarctic Mapping Tools for Matlab (Greene et al., 2017). (b) Profiles of atmospheric temperature (continuous line) and dew point temperature (dashed line) measured by the 10:24 UTC sounding from the AMF2 on the $4^{th}$ of January. Time-height cross sections of the reflectivity measured during the ascending (c) and descending (d) Cloudsat overpasses closest to McMurdo for the same day, indicated by the blue and red dashed lines in (a) (the colored circles in (a) and along the x-axis of (c) (d) are spaced by 30 s timesteps along the satellite path (equivalent to 228 km) and allow to better visualize the position of the cloud system). The vertical black dotted lines in (c) and (d) show the time of the closest approach for each overpass, and magenta lines indicate supercooled liquid water clouds as detected by CALIPSO.

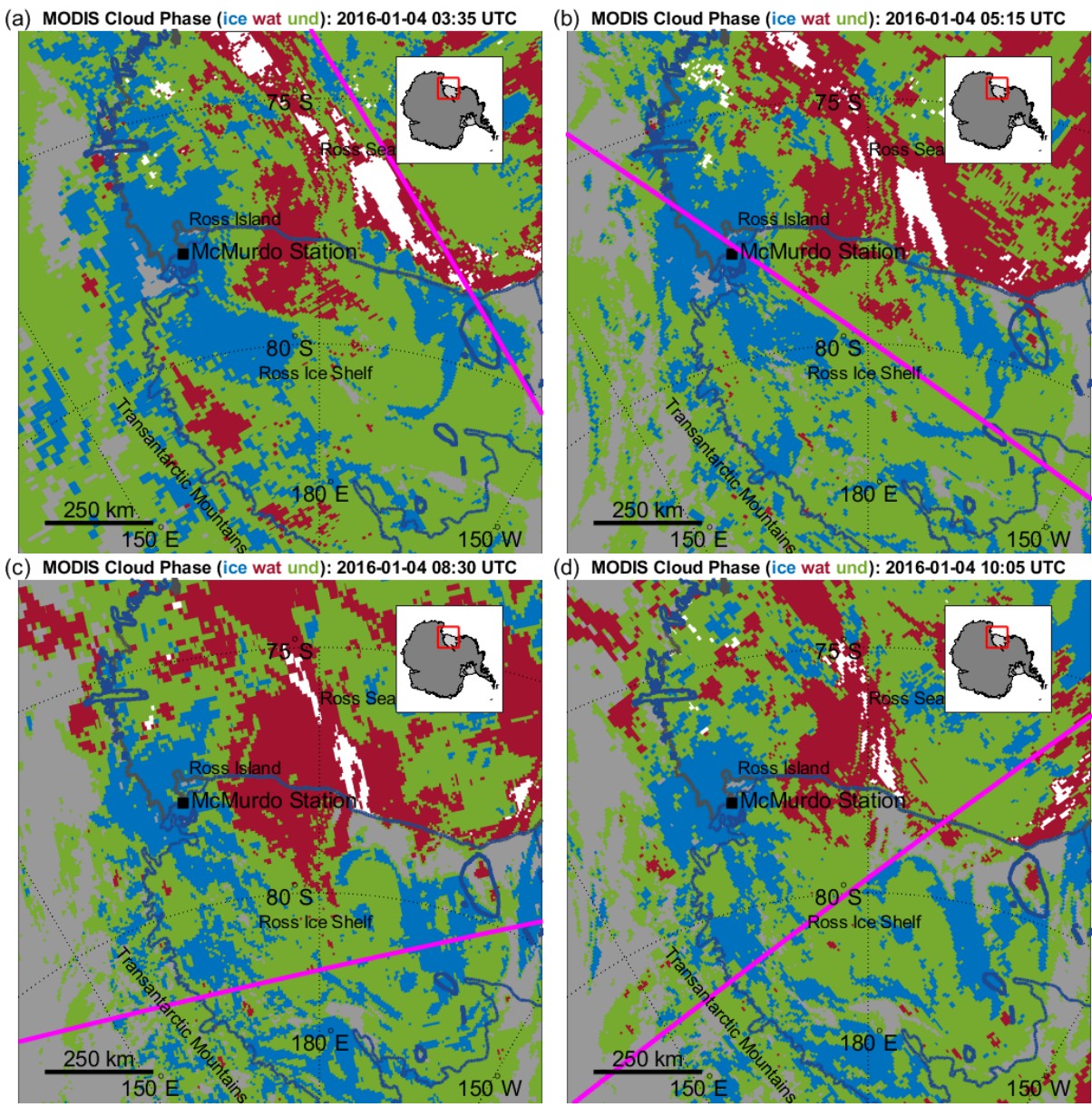

**Figure 3.** Time evolution of cloud top phase (ice, liquid or undetermined in blue, red and green, respectively) retrieved by subsequent MODIS overpasses on the $4^{th}$ of January at (a) 3:35, (b) 5:15, (c) 8:30 and (d) 10:05 UTC within the geographical area shown in Fig. 2a. The magenta lines correspond to the satellite ground track for each overpass.

### 4.2.1 Unusual dual-wavelengths ratios and intense riming

Of particular interest on the $4^{th}$ of January is the period between 07 and 12 UTC which is associated with a persistent supercooled liquid layer around 2 km AGL and characterised by clouds with the largest reflectivities (Fig. 4a). After applying the climatological relative calibrations determined for the whole AWARE field campaign (see Sect. 2.2), effective $DWR_{X,Ka}$ and $DWR_{Ka,W}$ are derived (Fig. 4b and 4c, respectively).

Microwave radiometer measurements were not available at McMurdo before the $29^{th}$ of January 2016. Before that date, liquid water path can still be roughly estimated thanks to multi-frequency radar observations (Tridon et al., 2020): using the Rayleigh plateau technique, Rayleigh reflectivity regions at cloud top can be identified (grey-shaded zones in Fig. 4c) and used to derive the two-way differential path-integrated attenuation ($\Delta$PIA, black thick line at the top of Fig. 4c associated with its own scale on the right axis in dB). While in general, $\Delta$PIA can be due to thick layers of supercooled liquid droplets or dense snow, the ice crystals in this case study are not expected to produce any significant attenuation. $\Delta$PIA can then be used to roughly estimate the supercooled liquid water path within this cloud (Tridon et al., 2020). Before 7:00 UTC and after 14:00 UTC, $\Delta$PIA is very close to 0 dB suggesting that ice water path (IWP) and liquid water path (LWP) are small and do not produce any detectable differential attenuation. During the period with largest $DWR_{Ka,W}$ (between 10:00 and 12:00 UTC), $\Delta$PIA reaches 0.25 dB. Assuming that only the supercooled liquid water contributes to the $\Delta$PIA, the corresponding LWP should be of the order of 100 $g\ m^{-2}$, according to recent refractive index models (Tridon et al., 2020). Note that $\Delta$PIA becomes slightly negative (-0.25 dB) between 9:30 and 10:30 UTC. This may be linked to the light snow shower reaching the ground around 9:00 UTC, and be explained by snow accumulating preferentially on the KAZR large flat radome. If this effect persists over the following hours, the true $\Delta$PIA between 10:00 and 12 UTC could be at most 0.5 dB, corresponding to a LWP of the order of 200 $g\ m^{-2}$.

In the upper part of the cloud (4 to 6 km AGL) where the temperature is between -25 and -40°C, the reflectivity and DWRs remain low. As aggregation in this temperature regime can be expected to be relatively weak, we expect plate-like particles and possibly polycrystals to dominate. Closer to the supercooled liquid layer, the $DWR_{X,Ka}$ is found only slightly enhanced (up to 5 dB, Fig. 4b) while the $DWR_{Ka,W}$ reaches the rather extreme value of 15 dB (Fig. 4c), i.e. the largest values in the AWARE field campaign data depicted in Fig. 1a. At temperatures lower than -15°C, aggregation is still expected to be limited and it is unlikely to be the process which leads to very large $DWR_{Ka,W}$ since $DWR_{X,Ka}$ would be also more strongly enhanced if the 15 dB were due to large aggregates. Conversely, such DWR combination strongly suggests an intense riming event, but this assumption cannot be readily corroborated by Doppler velocity because the entire period is affected by significant vertical motions.

### 4.2.2 Updrafts and mean fall velocity

The KAZR Doppler velocity, $V_D^{Ka}$ (positive when downward in our convention), is the result of the vertical air motion and the ice particle fall speed. For many instances of the case study, it features periods with negative values (Fig. 5a), i.e. updrafts. These updrafts are even more evident when exploiting KAZR Doppler spectra: the Doppler velocity of the slow edge of the

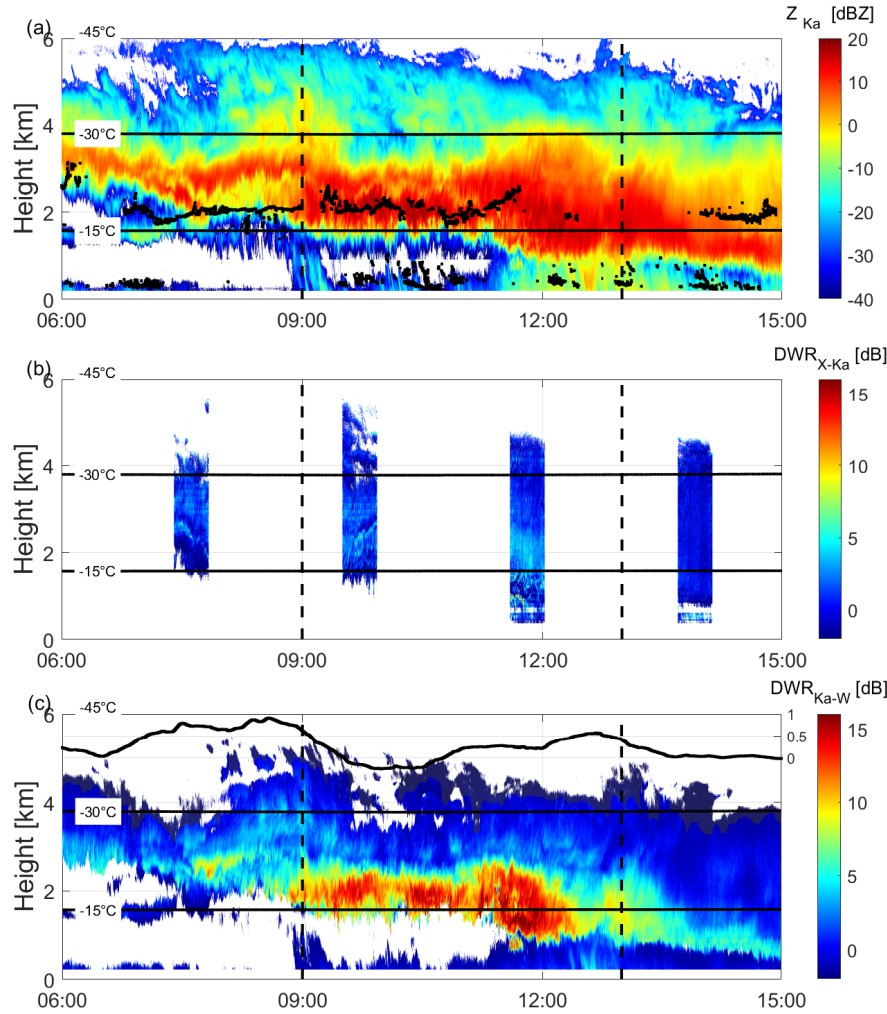

**Figure 4.** Time height cross section of the (a) KAZR reflectivity, (b) DWR$_{X,Ka}$, (c) DWR$_{Ka,W}$. The horizontal black lines indicate the -15 and -30°C levels while the vertical dashed lines delimit the period of large DWR$_{Ka,W}$ used to produce the density plots in Fig. 7, 8 and 9. The black dots in (a) show the liquid water as detected by the ARM high spectral resolution lidar (HSRL) cloud base height product (Silber et al., 2018b); see also the inset in Fig. 13a). The thick black curve in the upper part of (c) is the 2-way ΔPIA in dB (scale along the right axis) derived from Rayleigh scattering hydrometeors at cloud top (grey-shaded zones) following Tridon et al. (2020).

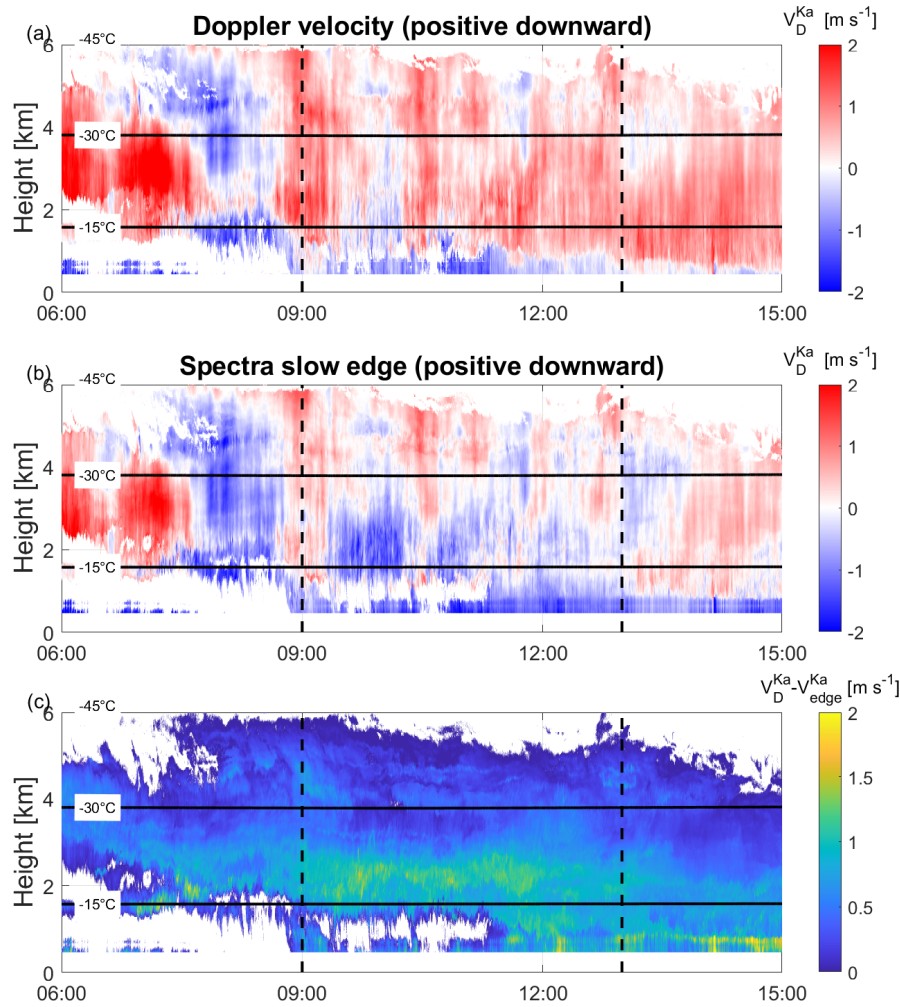

**Figure 5.** Time height cross section of the (a) KAZR Doppler velocity $V_D^{Ka}$, (b) Doppler velocity of KAZR spectra slow edge $V_{D,slowedge}^{Ka}$, (c) difference between KAZR Doppler velocity and Doppler spectra slow edge $V_D^{Ka} - V_{D,slowedge}^{Ka}$. The horizontal black lines indicate the -15 and -30°C levels while the vertical dashed lines delimit the period of large $DWR_{Ka,W}$ used to produce the density plots in Fig. 7, 8 and 9.

spectra $V_{D,slowedge}^{Ka}$ (i.e., the vertical velocity of the smallest hydrometeors detected in the sampling volume) is practically always negative (i.e. upward), and in some regions by almost $2\,\mathrm{m\,s}^{-1}$ (Fig. 5b).

During the period analysed, turbulence broadening is generally low (see Fig. 6a; further discussed in the following section). In such a case, subtracting the slow edge Doppler velocity from the Doppler velocity $V_D^{Ka} - V_{D,slowedge}^{Ka}$ gives a reasonable estimate of the reflectivity weighted mean fall velocity, or in other words, the Doppler velocity corrected from vertical air motion (Fig. 5c). Such estimate is however a lower limit because the smallest hydrometeors detected by the radar may not

have a negligible fall speed, an issue which will be exacerbated in regions of low radar sensitivity. In such instances, the actual updraft and, consequently, the derived mean fall velocity would both be underestimated. Nevertheless, such correction must be taken with caution because, in case of a high level of turbulence, it would lead to a large overestimation of the updraft and of the mean fall velocity.

Fig. 5a and b ($V_D^{Ka}$ and $V_{D,slowedge}^{Ka}$) clearly show vertical bands that alternate between variable saturation of blue/red colors typically due to alternating updrafts and downdrafts. After subtracting the slow edge Doppler velocity from the Doppler velocity $V_D^{Ka} - V_{D,slowedge}^{Ka}$ (Fig. 5c), there is no more abrupt change in time and the fall velocity gets larger towards the ground as expected for ice particles growing via deposition, riming or aggregation while they fall through a cloud. This suggests that the contribution of vertical air motions to the Doppler velocity has been correctly eliminated by exploiting the the slow edge of Doppler spectra.

The fall velocity of unrimed aggregates is known to be capped at around $1\,\mathrm{m\,s^{-1}}$, independent of their size because the increase of mass via aggregation is compensated by the enhanced drag due to the larger cross sectional area (Zawadzki et al., 2001; Kneifel and Moisseev, 2020). With values often reaching $1.4\,\mathrm{m\,s^{-1}}$ between 1.8 and $3\,\mathrm{km}$ AGL (Fig. 5c), the resulting mean fall velocity supports the presence of at least slightly rimed ice particles.

### 4.2.3   Limited turbulence and unusual spectral width signatures

Less directly, the spectral width $\sigma_D$ can be used to infer some information on ice properties as well (Maahn and Löhnert, 2017). The challenge is to separate the broadening due to the spread of hydrometeor fall velocities from the broadening due to air motion. For a vertically pointing cloud radar, the air motion broadening is mainly due to turbulence, wind shear and cross wind within the scattering volume (Borque et al., 2016). During this case study, the spectrum width observed by the KAZR (Fig. 6a) is mostly limited to rather small values compatible with the narrow spread of ice crystals fall velocity while only few layers with larger values are probably associated to gravity waves. Since the sampling volumes of KAZR and MWACR are very similar (due to the same range resolution and similar beam widths, i.e. 0.33 and 0.36°, respectively), air motion broadening should be identical for both radars. Any difference must be related to differential non-Rayleigh effects associated with large ice crystals when they are present, e.g. resulting in a narrower Doppler spectrum at the higher radar frequency. Interestingly, the MWACR spectral width is significantly larger than KAZR spectral width in the large $\mathrm{DWR}_{Ka,W}$ region (Fig. 6b), which leads to a negative differential spectral width between Ka and W bands $\delta\sigma_D^{Ka,W}$ (Fig. 6c). While the spectral width is expected to be larger at Ka-band in general, such a peculiar behaviour is possible for narrow size distributions of large ice crystals, as will be seen in Sect. 4.4.

In order to verify that the negative $\delta\sigma_D^{Ka,W}$ is not a spurious signal due to a possible mismatch of the radar beams, two-dimensional histograms (contoured frequency by altitude diagram, CFAD) of the differential Doppler velocity ($\delta V_D^{Ka,W}$) and $\delta\sigma_D^{Ka,W}$ for the period with high $\mathrm{DWR}_{Ka,W}$ are shown in Fig. 7. Above $3\,\mathrm{km}$ AGL, reflectivity, DWRs, and fall velocity are small (Fig. 4 and 5) suggesting that mostly small ice particles, thus Rayleigh scatterers, are present. In this case, both $\delta V_D^{Ka,W}$ and $\delta\sigma_D^{Ka,W}$ should be very close to zero. When non-Rayleigh targets are present, the Doppler velocity (still with a positive when downward convention) is generally smaller at the higher radar frequency because scattering effects reduce the backscatter

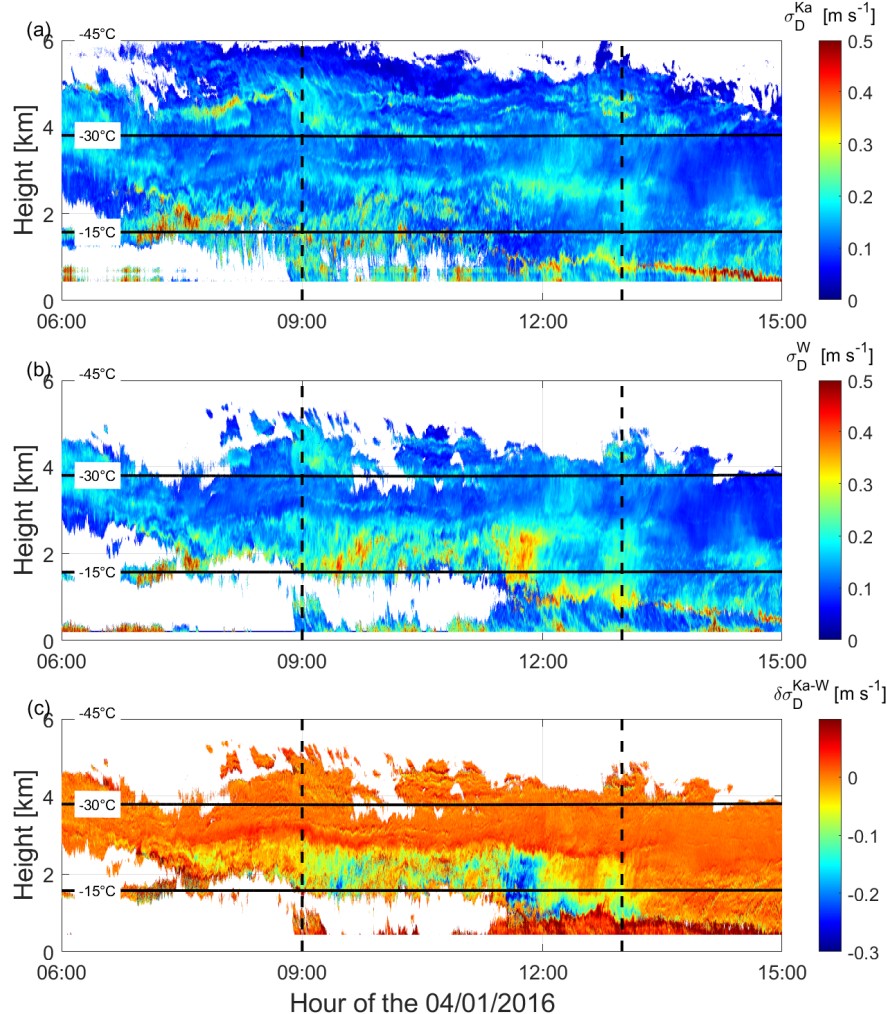

**Figure 6.** Time height cross section of the (a) KAZR spectral width $\sigma_D^{Ka}$, (b) MWACR spectral width $\sigma_D^W$ and (c) differential spectral width between KAZR and MWCAR $\delta\sigma_D^{Ka,W}$. The horizontal black lines indicate the -15 and -30°C levels while the vertical dashed lines delimit the period of large DWR$_{Ka,W}$ used to produce the density plots in Fig. 7, 8 and 9.

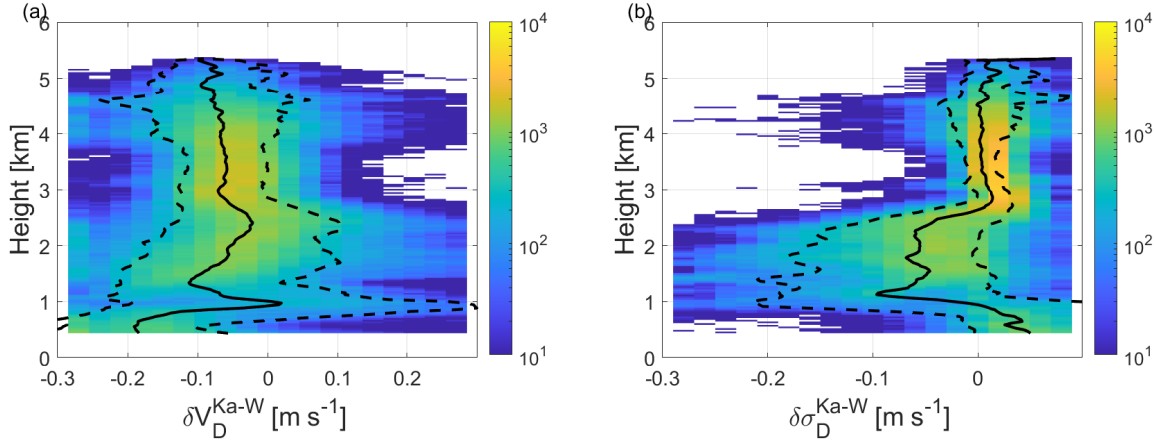

**Figure 7.** Contoured frequency by altitude diagram of the differential fall velocity $\delta V_D^{Ka,W}$ and differential spectral width $\delta \sigma_D^{Ka,W}$ between KAZR and MWACR during the period of high DWR$_{Ka,W}$ in between the vertical dash lines in Fig. 4c. The continuous and dashed black lines show the profiles of median, $10^{th}$ and $90^{th}$ percentiles of the distributions.

cross section of the largest and faster falling ice crystals. The slightly negative $\delta V_D^{Ka,W}$ (-0.5 m s$^{-1}$) — and hence larger $V_D^W$ — can only be explained by one of the radars slightly pointing off-zenith, as was found for the BAECC dataset (Kneifel et al., 2016). As a result, a small component of the horizontal winds is found along the pointing direction of the mispointing radar which explains the observed $\delta V_D^{Ka,W}$ difference. Conversely, being defined as the spread around the mean Doppler velocity,
the spectral width is not affected by a bias in Doppler velocity. As expected, $\delta \sigma_D^{Ka,W}$ is centered around zero for the small ice crystals present above 4 km AGL, confirming that the negative $\delta \sigma_D^{Ka,W}$ below 3 km AGL is not an artefact due to mispointing radar beams.

### 4.3 Qualitative evidence of narrow particle size distributions

The reflectivity observed by the three radars between 9:00 and 13:00 (the period delimited by the dashed lines in Fig. 4) are
combined in the so-called triple-frequency space (Fig. 8). The analysis is restricted to heights above 1 km and signal to noise ratio (SNR) larger than -5 dB, resulting in about 130000 data points (with resolution of 2 s by 30 m). The bullseye cluster centered around 0-0 dB corresponds to the upper part of the cloud (above 4 km) where the ice particles are small and nearly scatter in the Rayleigh regime at all frequencies. The cluster on the right hand side corresponds to the lower part of the cloud where supercooled liquid water is present. The combination of very high DWR$_{Ka,W}$ and rather small DWR$_{X,Ka}$ is known to
be the signature of rimed aggregates. In this case, this riming signature appears much stronger than previously observed, with the most frequent points being centered around DWR$_{X,Ka}$=3 dB and DWR$_{Ka,W}$=14 dB (magenta ellipse) while DWR$_{Ka,W}$ rarely exceeded 12 dB in previous triple-frequency radar field campaigns data sets (Kneifel et al., 2015; Mason et al., 2018; Dias Neto et al., 2019).

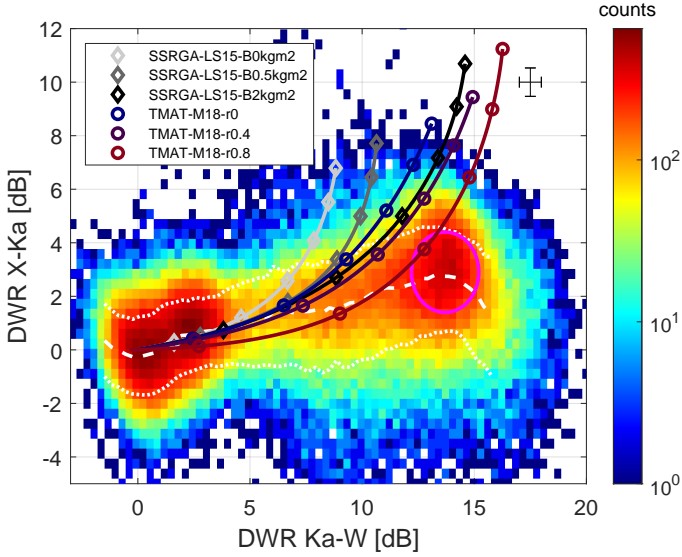

**Figure 8.** Two-dimensional histogram of measured dual-wavelength ratios (DWRs) for triple-frequency radar observations. The superimposed white dashed line (dotted lines) shows the median ($10^{th}$ and $90^{th}$ percentiles) of the histogram for the $DWR_{Ka,W}$ bins containing at least 500 points and the magenta ellipse highlights the unusual triple-frequency signature. The remaining superimposed lines represent the forward modeled DWR for an exponential distribution of particles with a mean mass diameter comprised between $0 < D_m \leq 6$ mm (each marker corresponding to 1 mm step) using various electromagnetic-microphysical models (see details in the text). The error bars in the top-right corner represent the intrinsic measurement uncertainties estimated for high SNR according to Doviak and Zrnic (2014).

For reference, the lines superimposed in Fig. 8 show DWRs of exponential distributions of unrimed and rimed ice crystals
forward modelled using various state-of-the-art scattering models designated as electromagnetic-microphysical (hereafter EM-MIC) models following the nomenclature introduced in Tridon et al. (2019). A much larger number of scattering models exists but here the focus is made on the most realistic ones. For example, models assuming pristine ice crystals with idealized shapes (such as spheres, rosettes or dendrites) are not shown. Light gray to dark gray lines correspond to the self-similar Rayleigh-Gans approximation (Hogan and Westbrook, 2014; Hogan et al., 2017) for realistic ensembles of ice aggregates (B model of
Leinonen and Szyrmer, 2015) with various degrees of riming (from unrimed SSRGA-LS15-B0kgm2 to heavily rimed SSRGA-LS15-B2kgm2). Note that the main difference between these SSRGA-LS15-Bxxx models is the quantity of supercooled water accreted to the ice particle, i.e. the degree of riming, but during the generation of the ice aggregates, depositional growth and aggregation are intrinsically included by the explicit simulation of the aggregation of monomers of various sizes. Blue, purple and red lines correspond to T-matrix scattering calculations for oblate spheroid ice crystals with an axial ratio of 0.6 and
composed of a homogeneous ice-air mixture (soft spheroid model). The different lines correspond to various densities which are determined from the density factor $r$ introduced by Mason et al. (2018) (from unrimed TMAT-M18-r0 when $r = 0$ to nearly hail TMAT-M18-r0.8 when $r = 0.8$).

During the four hours of observations merged in Fig. 8, a variety of microphysic processes are certainly occurring, and due to the complexity of ice particles and ice PSD shapes, there is a large natural variability in the observations. On the contrary, the scattering models can only represent an average behavior of a mixture of ice habits. The objective here is to find the scattering model which best match the observations on average and, hence, to detect, from the observed reflectivity signatures, the fingerprints of the dominant microphysical process in shaping the ice particles. However, multi-frequency radar observations are inevitably noisy not only because of the possible radar volume mismatch, but also because of the intrinsic noisiness of radar measurements which decreases with increasing SNR (Doviak and Zrnic, 2014). For the configuration of the ARM radars during AWARE, $\text{DWR}_{X,Ka}$ and $\text{DWR}_{Ka,W}$ individual observations are associated with an uncertainty of around 0.5 dB at high SNR (as illustrated by the error bars in the top-right corner of Fig. 8). Nevertheless, the median and $10^{th}$ and $90^{th}$ percentiles of the density plot as function of $\text{DWR}_{Ka,W}$ (white lines in Fig. 8) highlight the average trend and natural variability of the observations which can then be compared to the theoretical model lines. At low $\text{DWR}_{Ka,W}$, all the scattering models shown in Fig. 8 are in agreement with the trend of the observations. But at large $\text{DWR}_{Ka,W}$, Fig. 8 clearly shows that, if exponential distributions are assumed, even the scattering models corresponding to rimed particles deviate significantly from the median of the distribution, and hence none of the selected scattering models seems to convincingly explain the observed unusual triple-frequency signatures.

Mason et al. (2019) showed that the shape of the ice particle size distribution (PSD) can also affect the triple-frequency radar signature: reducing the width of a PSD (e.g. by increasing the shape parameter $\mu$ of a gamma PSD) has a similar effect to that of increasing the particle density. As a result and since riming has been shown to be correlated with narrow size distributions (Garrett et al., 2015), a narrow PSD could amplify the triple frequency signature of riming and is the most plausible way to explain the extreme signature observed in this case.

### 4.4 Constraining ice particle properties from multi-frequency radar observations

In order to take into account the effect of the shape of the PSD (see examples in Fig. 12a), we consider gamma distributions of the form:

$$N(D) = N_0^* f(\mu)(D/D_m)^\mu \exp^{-\Lambda D} \tag{1}$$

where $D$ is the maximum dimension of ice particles, $\Lambda$ and $\mu$ are the slope and shape parameters, $D_m = (1 + \mu + b_m)/\Lambda$ is the mean mass diameter, $b_m$ is the exponent of the mass-size relation associated to the EM-MIC model and $f(\mu)$ is a normalisation factor following Testud et al. (2001). In comparison to an exponential distribution, a gamma distribution (with $\mu$ larger than zero) leads to larger $\text{DWR}_{Ka,W}$ for any EM-MIC model (Mason et al., 2019, and Fig. 9a). However, it also leads to a reduced spectral width and, as was shown in Sect. 4.2, the observed negative $\delta\sigma_D^{Ka,W}$ is a very specific feature and is an evidence that the PSD is wide enough to contribute to the spectral width. Therefore, combining the triple-frequency radar signature with the observed $\sigma_D$s offers a way to constrain the best EM-MIC model matching the observations (e.g., by comparing the average trend of the density plots of these observations as function of the $\text{DWR}_{Ka,W}$ to the theoretical lines provided by the EM-MIC models such as in Fig. 8). Furthermore, even if the estimation of $V_D - V_{D,slowedge}$ requires a negligible turbulence

broadening and a high radar sensitivity, Sect. 4.2 and Fig. 6 suggest that these conditions are reasonably fulfilled for this case study. Then, since the $V_D - V_{D,slowedge}$ parameter is unaffected by the vertical wind, it is a further parameter that can be used to evaluate EM-MIC models, contrary to the Doppler velocity. To this aim, Doppler spectra with a realistic noise level are simulated following the methodology described in Tridon and Battaglia (2015) and $v_{D,slowedge}$ is determined as being the Doppler velocity of the first bin with spectral reflectivity larger than the noise, as for the observed Doppler spectra (see examples of simulated and observed Doppler spectra in Fig. 12b and c).

The resulting density plots of observations are shown in Fig. 9, in which the superimposed theoretical lines correspond to a new EM-MIC model briefly introduced below. By using a similar methodology, Fig. S1 and Fig. S2 of the supplementary material provide an assessment on how well the two types of rimed aggregates EM-MIC models discussed in the previous section (i.e. SSRGA-LS15-B1kgm2 and TMAT-M18-r0.4) fit the radar observations. Interestingly, despite choosing the most adequate degree of riming, these models appear to be inconsistent with the measurements for the following reasons:

- For the SSRGA-LS15 type (Fig. S1), a rather high degree of riming (equivalent liquid water path of 1 kg m$^{-2}$) is required to produce large enough DWR$_{Ka,W}$. This is very surprising because this EM-MIC model corresponds to heavily rimed particles while the small $\Delta$PIA between Ka and W-bands (see Sect. 4.2) suggests that the amount of observed supercooled liquid water is relatively small. This leads to excessive simulated fall velocities and spectral widths, with particularly high $V_D^{Ka} - V_{D,slowedge}^{Ka}$ at small DWR$_{Ka,W}$, resulting in a completely inadequate sloping of this parameter with increasing DWR$_{Ka,W}$.

- A much better agreement is found for the TMAT-M18 type when using a density factor r=0.4 and shape parameter $\mu = 4$ (Fig. S2). Nevertheless, the resulting $V_D^{Ka} - V_{D,slowedge}^{Ka}$ and $\sigma_D^{Ka}$ are slightly too large. Furthermore, this model appears less physical since it suggests that very high DWR$_{Ka,W}$ (larger than 20 dB) could be reached in case of very narrow size distributions ($\mu \geq 16$), while the observations suggest a clear cut-off above 15 dB.

Note that the Doppler velocity comparison is not without uncertainty: first, the proposed vertical air motion correction is an approximation; and second, the hydrodynamic theory for ice particles of complex shape is still a topic of active research, with the different hydrodynamic models proposed in the literature (Böhm, 1992; Khvorostyanov and Curry, 2005; Heymsfield and Westbrook, 2010) known to lead to slightly different fall velocities (different line widths in Fig. 9b). Nevertheless, this uncertainty is much smaller than the large overestimation found with the SSRGA-LS15-B1kgm2 model.

A possible explanation for the excessive Doppler velocities of the SSRGA-LS15-B1kgm2 models could be in its mass-size relation. Indeed, the mass-size relation parameters of the Leinonen and Szyrmer (2015) B model for heavily rimed particles are quite different from those derived from various observations according to the comprehensive review made by Mason et al. (2018, their Fig. 1). The rather large prefactor with low exponent lead to particularly large masses for corresponding sizes. While the aggregation and riming model used in Leinonen and Szyrmer (2015) is widely accepted and provides physically reasonable particle shapes (Seifert et al., 2019; Karrer et al., 2020), it remains debatable how realistic it is to cluster the ice particles by the equivalent LWP of supercooled droplets through which they sediment. This model indeed assumes an ideal seeder-feeder cloud situation with the same riming efficiency at all ice particles sizes.

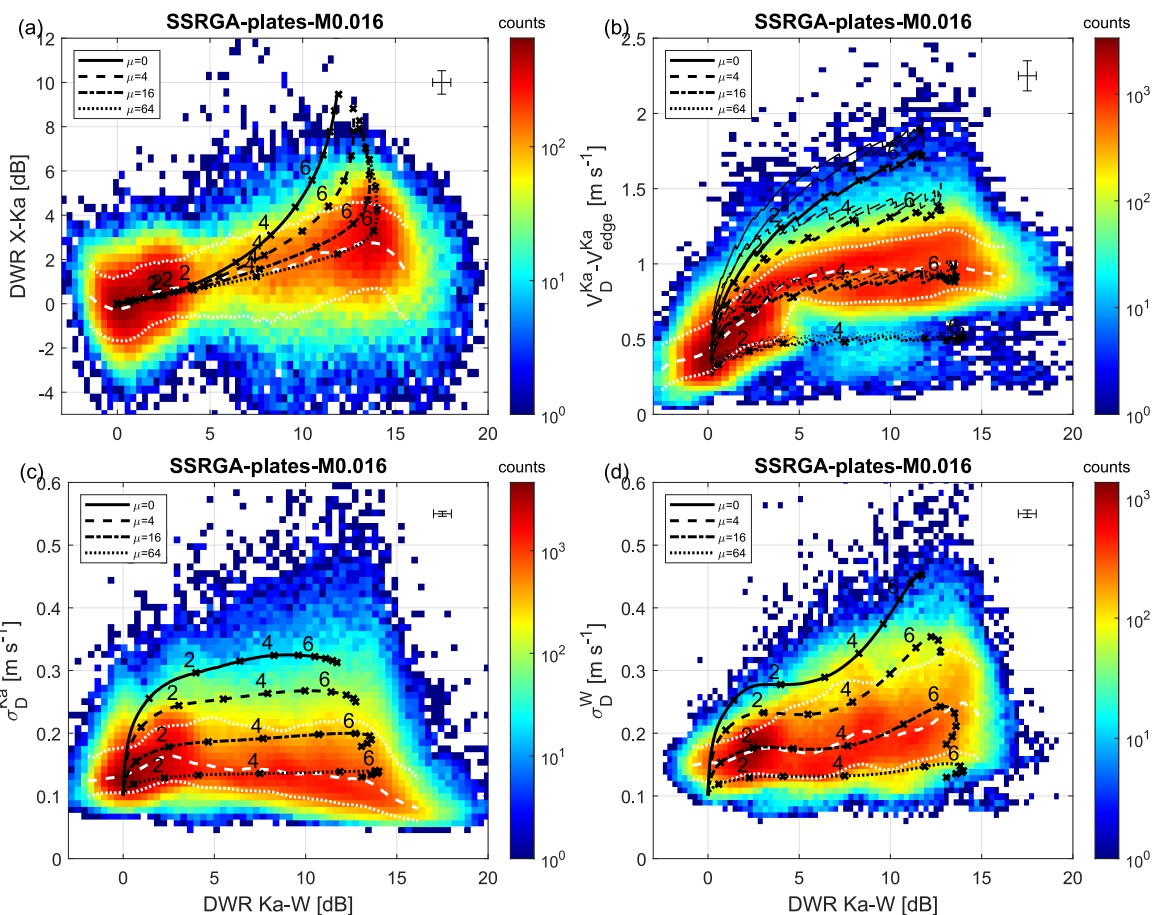

**Figure 9.** Two-dimensional histograms of observed (a) $DWR_{X,Ka}$, (b) difference between Ka-band Doppler velocity and Doppler spectra slow edge $V_D^{Ka} - V_{D,slowedge}^{Ka}$, (c) Ka-band spectral width $\sigma_D^{Ka}$ and (d) W-band spectral width $\sigma_D^W$ as function of observed $DWR_{Ka,W}$. The superimposed white dashed line (dotted lines) shows the median ($10^{th}$ and $90^{th}$ percentiles) of the histogram for the $DWR_{Ka,W}$ bins containing at least 500 points. The superimposed black lines represent the corresponding parameters forward modeled with the SSRGA for a gamma distribution of aggregates of plates with various $\mu$ (see the legends in the plots), a mean mass diameter of $0 < D_m \leq 10$ mm (each marker corresponding to 1 mm step) and a normalized rimed mass $\mathcal{M} = 0.016$ (see the text for details). The error bars in the top-right corner of each panel represent the intrinsic measurement uncertainties estimated for high signal to noise ratios according to Doviak and Zrnic (2014). In panel (b), thick lines correspond to calculations of fall velocities using the Böhm (1992) model while thin lines correspond to the Khvorostyanov and Curry (2005) and Heymsfield and Westbrook (2010) models.

**Table 2.** Parameters of the main EM-MIC models used in this study: mass-size relation parameters (prefactor $a_m$ and exponent $b_m$), effective aspect ratio ($\alpha_{eff}$) and SSRGA parameters (kurtosis parameter $\kappa$, power-law prefactor $\beta$ and exponent $\gamma$, and correction factor $\zeta_1$) averaged over the 1-10 mm range of sizes. To obtain the parameters of the other EM-MIC models of Fig. 8, the reader can refer to Leinonen and Szyrmer (2015) and Mason et al. (2018) for the SSRGA-LS15 and TMAT-M18 series, respectively.

| EM-MIC model | $a_m$ | $b_m$ | $\alpha_{eff}$ | $\overline{\kappa}$ | $\overline{\beta}$ | $\overline{\gamma}$ | $\overline{\zeta_1}$ |
|---|---|---|---|---|---|---|---|
| TMAT-M18-r0.4 | 0.68 | 2.34 | 0.6 | N/A | N/A | N/A | N/A |
| SSRGA-LS15-B1kgm2 | 0.37 | 2.11 | 0.79 | 0.17 | 2.55 | 3.34 | 0.06 |
| SSRGA-plates-$\mathcal{M}$0.016 | 0.52 | 2.43 | 0.53 | 0.26 | 2.05 | 2.57 | 0.07 |

Instead, we propose to cluster rimed particles according to their normalized rime mass, $\mathcal{M}$, defined by Seifert et al. (2019) as the ratio of the particle mass $m$ to the mass $m_g$ of a $700\ \mathrm{kg\,m^{-3}}$ graupel of equivalent size. Such parameter is better suited to represent successive stages of riming since its definition literally translates the asymptotic increase of $m$ toward $m_g$, and it allowed Seifert et al. (2019) to illustrate the self-similarity of the conversion of aggregates to graupel-like particle. In order to build new EM-MIC models corresponding to specific $\mathcal{M}$ values, we used the open code provided by Leinonen and Szyrmer

(2015) and produced an ensemble of rimed aggregates with a wide range of $\mathcal{M}$ (leading to more diversified degrees of riming than what was proposed in Leinonen and Szyrmer, 2015). Since the DWRs observed during the case study start to increase in the plate-like growth regime, plate aggregates or polycrystals are quite likely and we chose plates as primary ice crystal shape for the simulations. We then used the Snowscatt tool (Ori et al., 2020) to derive the SSRGA parameters for various $\mathcal{M}$ classes. While the full discussion of the resulting rime ice particle classes is beyond the scope of this study and will be fully described

in a subsequent paper, the resulting $\mathcal{M}$ classes provide mass-size parameters which are significantly different from previous studies, but are consistently increasing (not shown) in agreement with riming theory, in particular, with an exponent increasing from 2 (fractal geometry of unrimed aggregates) to 3 (spherical particles, i.e. fully rimed). Especially, for corresponding sizes, these mass-size parameters lead to smaller masses and fall velocities than the Leinonen and Szyrmer (2015) B model. With a normalized rimed mass $\mathcal{M} = 0.016$ and a $\mu$ value on the order of 16, the resulting slightly rimed particle class provides a

reasonable agreement with the observed triple-frequency DWRs, spectral widths, and fall velocity, all at the same time (Fig. 9). In this scenario, the high $\mathrm{DWR}_{Ka,W}$ cluster in Fig. 9a corresponds to mean mass diameter $D_m$ ranging from 5 to 7 mm. For comparison, the parameters of the SSRGA-plates-$\mathcal{M}$0.016 model are compared to those of previous EM-MIC models in the Table 2. Apart from the mass-size parameters, a significant difference resides in the effective aspect ratio ($\alpha_{eff}$): for the SSRGA-LS15-B1kgm2 model, it is closer to unity suggesting rounded particles and heavy riming, while SSRGA-plates-

$\mathcal{M}$0.016 has a value closer to 0.6, the value widely accepted in the literature and more consistent with slight riming.

## 4.5   Retrieval of ice properties

In order to constrain the ice particle properties from multi-frequency radar observations, numerous assumptions are required. The most important one is the EM-MIC model (and its associated mass-size relation) chosen to describe the type of ice

particles. In an effort to evaluate the uncertainty associated with this choice, a simple retrieval of ice properties using the multi-frequency radar data and the most likely ice particle types is proposed in this section. The objective here is to find whether an EM-MIC model can describe the triple-frequency radar observations of this case in a physically consistent way. A fully realistic retrieval of the microphysics properties is out of the scope of this study and would require a methodology based on an ice model providing continuous description of ice properties and scattering cross sections as function of the degree of riming similarly to what was proposed by Leinonen et al. (2018) and Mason et al. (2018).

The retrieval assumes an EM-MIC model and is applied for each ice particle type independently. Its aim is to retrieve parameters such as ice particle number concentration $n_i$, ice water content (IWC) and mean mass diameter $D_m$. To do so, it is required to invert the ice particle size distribution (PSD). By assuming a gamma PSD (Eq. (1)), three parameters must be retrieved: $N_0^*$, $\mu$ and $D_m$. It has been shown that the use of a maximum ice particle size $D_{max}$ has a significant impact on radar retrievals (Gergely, 2019). Nevertheless, because we want to mainly focus here on the impact of the choice of the EM-MIC model, we decided to fix $D_{max}$ to 3 times $D_m$, following Szyrmer and Zawadzki (2014). The core of the methodology is to retrieve $\mu$ and $D_m$ via a simple minimization technique (such as in Turk et al., 2011) and its main steps are as follows:

1. For each EM-MIC candidate, we build a multidimensional lookup table that provides the forward simulated $\mathrm{DWR}_{X,Ka}^{sim}$, $\mathrm{DWR}_{Ka,W}^{sim}$ and $V_D^{sim} - V_{D,slowedge}^{sim}$ at the Ka band (where the Ka subscript is omitted for simplicity) corresponding to any $\mu - D_m$ pair.

2. For each combination of measurements, $\mathrm{DWR}_{X,Ka}^{obs}$, $\mathrm{DWR}_{Ka,W}^{obs}$ and $V_D^{obs} - V_{D,slowedge}^{obs}$, the best matching $\mu - D_m$ pair is found by minimizing the cost function CF:

$$\mathrm{CF} = \frac{\left|\mathrm{DWR}_{X,Ka}^{obs} - \mathrm{DWR}_{X,Ka}^{sim}\right|}{\sigma_{\mathrm{DWR}_{X,Ka}}} + \frac{\left|\mathrm{DWR}_{Ka,W}^{obs} - \mathrm{DWR}_{Ka,W}^{sim}\right|}{\sigma_{\mathrm{DWR}_{Ka,W}}} + \frac{\left|(V_D^{obs} - V_{D,slowedge}^{obs}) - (V_D^{sim} - V_{D,slowedge}^{sim})\right|}{\sigma_{V_D - V_{D,slowedge}}}, \quad (2)$$

where $\sigma_{\mathrm{DWR}_{X,Ka}}$, $\sigma_{\mathrm{DWR}_{Ka,W}}$ and $\sigma_{V_D - V_{D,slowedge}}$ represent the sum of the measurements uncertainties and natural variability of these observations. Based on the joint histograms of Fig. 9, these uncertainties have been set to 3 dB, 1.5 dB and 0.2 m s$^{-1}$, respectively. This provides a direct mapping from a set of measurements $\mathrm{DWR}_{X,Ka}^{sim}$, $\mathrm{DWR}_{Ka,W}^{sim}$ and $V_D^{sim} - V_{D,slowedge}^{sim}$ to the unknowns $\mu - D_m$.

3. The mapping from the measurements to the unknowns is highly non-linear. The uncertainty associated to this variability is taken into account via Monte Carlo propagation. Namely, the retrieval is performed several times on an ensemble obtained by perturbing each measurement via normally distributed measurement uncertainties with standard deviations $\sigma_{\mathrm{DWR}_{X,Ka}}$, $\sigma_{\mathrm{DWR}_{Ka,W}}$ and $\sigma_{V_D - V_{D,slowedge}}$, respectively. For each set of measurements, a $\mu - D_m$ pair is retrieved and the resulting retrieval uncertainties $\sigma_\mu$ and $\sigma_{D_m}$ are obtained by taking the standard deviation of the ensemble of retrieved $\mu$ and $D_m$ values.

4. Once $\mu$ and $D_m$ and their uncertainties are retrieved for a data voxel, $N_0^*$ can be directly derived from the observed reflectivity $Z_{Ka}$ thanks to the relation

$$N_0^* = \frac{\pi^5 |K|^2}{\lambda_{Ka}^4} \frac{Z_{Ka}}{\int \sigma_b^{\text{EM-MIC}}(D) D^\mu \exp^{-\Lambda D} dD}, \quad (3)$$

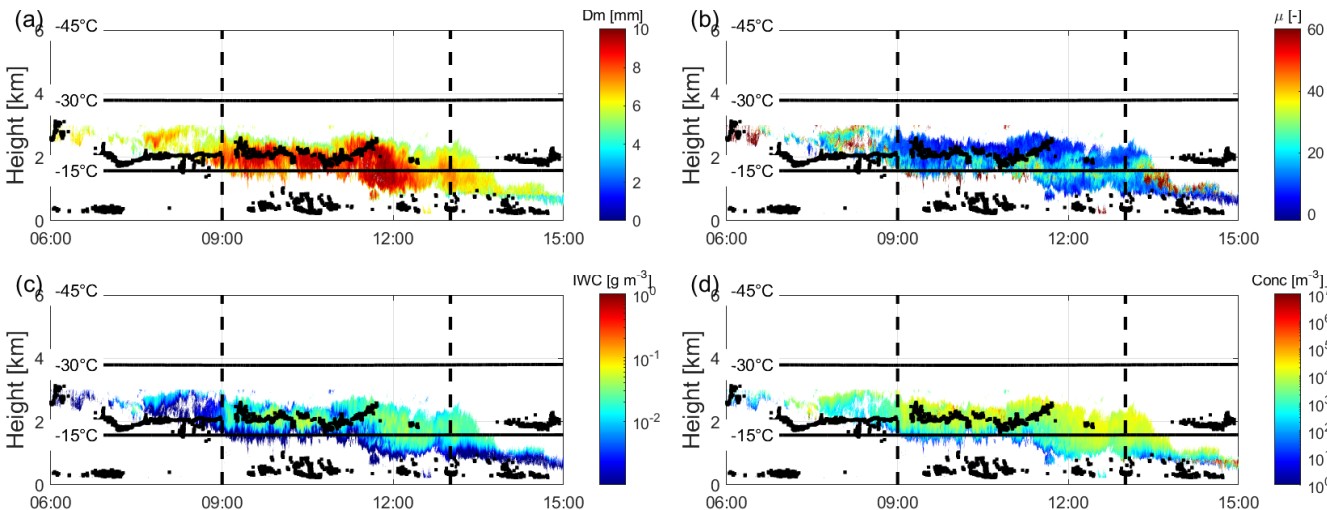

**Figure 10.** Time height cross section of the retrieved (a) $D_m$, (b) $\mu$, (c) IWC and (d) $N_i$ when using the SSRGA-plates-$\mathcal{M}0.016$ model, for all pixels where $\mathrm{DWR}_{Ka,W}$ is larger than 4 dB. The black dots show the location of supercooled liquid water as detected by the ARM high spectral resolution lidar.

while its standard deviation $\sigma_{N_0^*}$ is computed via error propagation, assuming that $\sigma_\mu$ and $\sigma_{D_m}$ are independent.

5. Finally, IWC and ice number concentration $n_i$ and their uncertainties are computed using:

$$n_i = N_0^* \int D^\mu \exp^{-\Lambda D} dD, \tag{4}$$

$$IWC = N_0^* \int a_m D^{b_m} D^\mu \exp^{-\Lambda D} dD, \tag{5}$$

where $a_m$ and $b_m$ are the prefactor and exponent of the mass-size relation associated to the EM-MIC model.

The retrieval is applied with the three EM-MIC models, which were found to better describe the joint histograms of observations in Fig. 9, Fig. S1 and Fig. S2: SSRGA-plates-$\mathcal{M}0.016$, SSRGA-LS15-B1kgm2 and TMAT-M18-r0.4. As an example, Fig. 10 illustrates the results of the retrieval using the SSRGA-plates-$\mathcal{M}0.016$ model. For simplicity, it is applied only to the data identified as rimed by requiring the $\mathrm{DWR}_{Ka,W}$ to be larger than 4 dB (of course, such a simple threshold is valid for

this particular case study only because aggregation is negligible in the unrimed part of the cloud). Apart from $\mu$, which appears slightly noisy, the retrieved fields are reasonably homogeneous, i.e. there is a good spatial coherence. Even though the retrieved $\mu$ values may appear anomalously large, the retrieved $\Lambda$ (not shown) range between 1 and 5 $\mathrm{mm}^{-1}$, consistently with previous studies (Brandes et al., 2007; Gergely, 2019). From 2.6 to 2.3 km AGL, $D_m$ strongly increases towards the ground, highlighting the layer where riming is the most efficient and the probable top of the supercooled liquid layer. From 2 km AGL

downward, there is a clear decrease of $n_i$ and IWC, consistent with sublimation of snow when it mixes with dry air in the boundary layer.

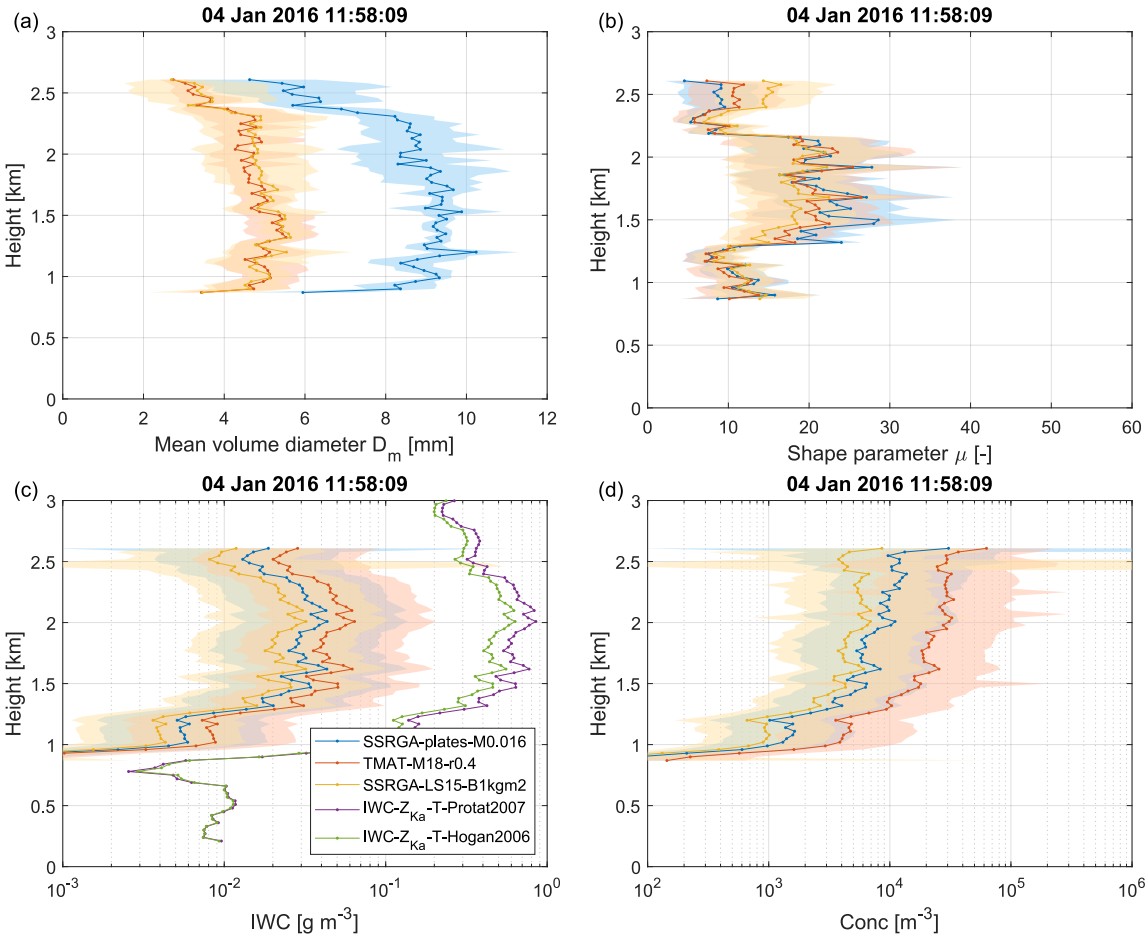

**Figure 11.** Comparison of (a) $D_m$, (b) $\mu$, (c) IWC and (d) $n_i$ profiles at 11:58:09 UTC retrieved with the most probable EM-MIC models identified in Sect. 4.4. Colored shadings show the corresponding uncertainties obtained via Monte Carlo propagation as described in the retrieval methodology.

The fields of parameters retrieved with SSRGA-LS15-B1kgm2 and TMAT-M18-r0.4 (not shown) are similarly homogeneous as for SSRGA-plates-$\mathcal{M}$0.016. However, significant and consistent differences are found throughout the entire case study. For simplicity, the retrieved parameters are compared in Fig. 11 for a single profile at 11:58:09 UTC where the DWR$_{Ka,W}$ is maximum, but the results are similar for all other profiles. While all three EM-MIC models agree fairly well on the retrieved $\mu$ values peaking at 20 between 1.4 and 2.2 km AGL, the range of $D_m$ values is not very well constrained owing to the discrepancies between the EM-MIC model attributes (Table 2): SSRGA-plates-$\mathcal{M}$0.016 suggests $D_m$ values about twice as large as SSRGA-LS15-B1kgm2. Sensitivity tests (not shown) indicate that this large difference is mainly due to the unexpectedly high aspect ratio associated with the SSRGA-LS15-B1kgm2 model. With a smaller and more realistic aspect ratio, the particles of the SSRGA-plates-$\mathcal{M}$0.016 model have a shorter dimension along the scattering direction, resulting in lesser destructive interferences at higher radar frequencies, a weaker reduction of the backscatter cross section of large particles, and hence, smaller DWRs for corresponding sizes. With TMAT-M18-r0.4, this effect is cancelled by the distinct method used for the scattering calculations and the resulting $D_m$ are close to those of SSRGA-LS15-B1kgm2 by coincidence. In a nutshell, the resulting uncertainty on $D_m$ is large because the particle shape is under-constrained. We can only conclude with confidence that the particles are large, with a $D_m$ on the order of 4 mm or larger. Furthermore, knowing that both the SSRGA-plates-$\mathcal{M}$0.016 and SSRGA-LS15-B1kgm2 models have been derived from the same aggregation and riming model, additional work is needed to determine which aspect ratio is the most realistic and to be able to derive $D_m$ with a better accuracy. Interestingly, $n_i$ and IWC retrieved with the three different EM-MIC models are in a fairly good agreement despite the large difference in $D_m$, which suggests that these parameters are rather well constrained when combining reflectivity observed at X, Ka and W-bands and the $V_D^{sim} - V_{D,slowedge}^{sim}$ estimate. Likewise, figure 11c also shows the IWC retrieved using the climatological IWC-$Z_{Ka}$-T relationships of Hogan et al. (2006) and Protat et al. (2007, for midlatitude). They result in IWCs more than one order of magnitude larger than the EM-MIC models suggesting that such simple statistical relations are not suited for specific conditions like the narrow size distributions of this case.

The three EM-MIC models consistently suggest that the unusual triple-frequency radar signatures were due to narrow PSDs. Even though the retrieved shape parameter values $\mu = 20$ are in the high end or above the range commonly found in the literature (Brandes et al., 2007; Tiira et al., 2016; Mason et al., 2019), such PSDs have a sensible shape (Fig. 12a, green line) and the corresponding foward-modeled Doppler spectra (Fig. 12b, thick lines) are comparable to the observed ones (Fig. 12c). When an exponential PSD is used, the forward-modeled Doppler spectra are considerably wider than in the observations, leading to spectral widths much larger than those predominantly observed (see Fig. 9c and d). In order to obtain Doppler spectra as narrow as those observed, with an exponential PSD and the SSRGA-plates-$\mathcal{M}$0.016 model (or the TMAT-M18-r0.4 model, see Fig. S2), a $D_m$ smaller than 1 mm is required, which is then not compatible with the large DWRs observed. This further confirms that narrow PSDs are required to explain the observed triple-frequency signatures.

In summary, despite the lack of in-situ observations for constraining the ice particles properties, the detailed exploitation of triple-frequency radar observations allowed us to conclude that the specific radar signature observed during this case study was due to narrow distributions of large and slightly rimed plate polycrystals. It the next section, we devise a bin model experiment to better understand whether riming and such a narrow PSD can be physically consistent.

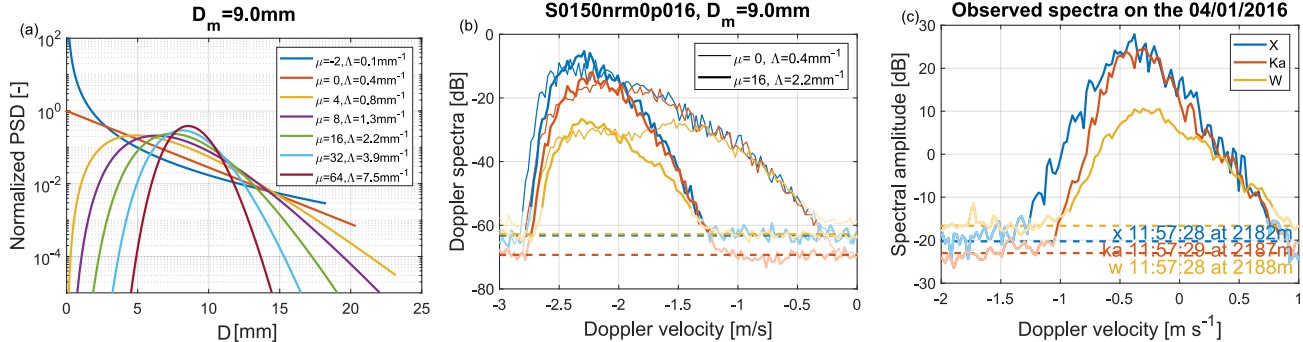

**Figure 12.** (a) Gamma PSDs with $D_m$ of 9 mm and various shape parameters $\mu$ (see the legend for numeric values) as function of the maximum dimension of ice particles. (b) Corresponding Doppler spectra at X, Ka and W-band (see color key in panel (c)) forward-modeled using the SSRGA-plates-$\mathcal{M}$0.016 model, for an exponential PSD ($\mu = 0$, thin line) and a narrow PSD ($\mu = 16$, thick line). (c) Examples of observed Doppler spectra where similar $D_m$ and $\mu$ are suggested by the retrieval using the SSRGA-plates-$\mathcal{M}$0.016 model (see Fig. 11). In (b) and (c), the horizontal dashed lines represent the noise level estimated from each Doppler spectra and the portions of lines with weak color saturation highlight the parts of the Doppler spectra which are identified as noise.

## 4.6 Bin model experiment: can a plausible scenario reproduce the observed narrow PSD of rimed ice particles?

In-situ airborne observations of ice PSDs over Antarctica are relatively scarce and are commonly performed using instruments such as the Cloud Particle Imager (CPI Lawson et al., 2001) and the Cloud Imaging Probe (CIP Lachlan-Cope et al., 2016),
which are limited to a particle maximum dimension of $\sim$1.5 mm. This lack of a comprehensive observational database of Antarctic ice precipitation PSDs combined with instrument-detectable particle size limitations inhibits any pertinent comparison with the results presented here of ice particles with sizes on the order of a few to several mm generating the observed triple-frequency radar signatures. Moreover, the elevated altitude range characteristic of these radar signatures further impedes comparisons to ice particle properties derived from Antarctic ground-based observations, which are strongly influenced by
low-level ice sublimation (e.g., Grazioli et al., 2017b) and blowing snow events (e.g., Loeb and Kennedy, 2021), frequently occurring over the region. Instead, we have performed a modeling exercise in order to establish whether a plausible riming scenario based on AWARE observations could develop the detected triple-frequency signature, thereby adding a physical context to this analysis, which exemplifies that such narrow ice PSDs are realistic.

### 4.6.1 Model setup and initialization

To examine whether a plausible riming scenario could develop the triple-frequency signature detected in the observations, we use the Distributed Hydrodynamic Aerosol and Radiative Modeling Application (DHARMA) model (Stevens et al., 2002) coupled with the Community Aerosol-Radiation-Microphysics Application (CARMA) size-resolved bin microphysics model (Ackerman et al., 1995; Jensen et al., 1998). Our main hypothesis in this modeling exercise is that the high $\mu$ values suggested

by the observations are most likely to occur if the ice hydrometeors dominating the radar echoes originate from a shallow generating layer and experience little to no mixing before the combined $DWR_{X,Ka}$ and $DWR_{Ka,W}$ signatures are produced. That is because a deep generating layer or strong mixing of rimed ice hydrometeors would necessarily lead to stronger dispersion of size-dependent ice particle fall velocities at given air volumes and broadening of the ice PSD (lower $\mu$). Such broadening of the PSD implies larger contribution of particle sizes producing lower $DWR_{Ka,W}$ and/or higher $DWR_{X,Ka}$ (Fig. 9) to the total output signatures resulting in deviation from the observed DWR values. This hypothesis is supported by the small spectral width values (Fig. 6a) and the largely stable atmospheric profile during the event indicated by the potential temperature sounding measurements (Fig. 13a).

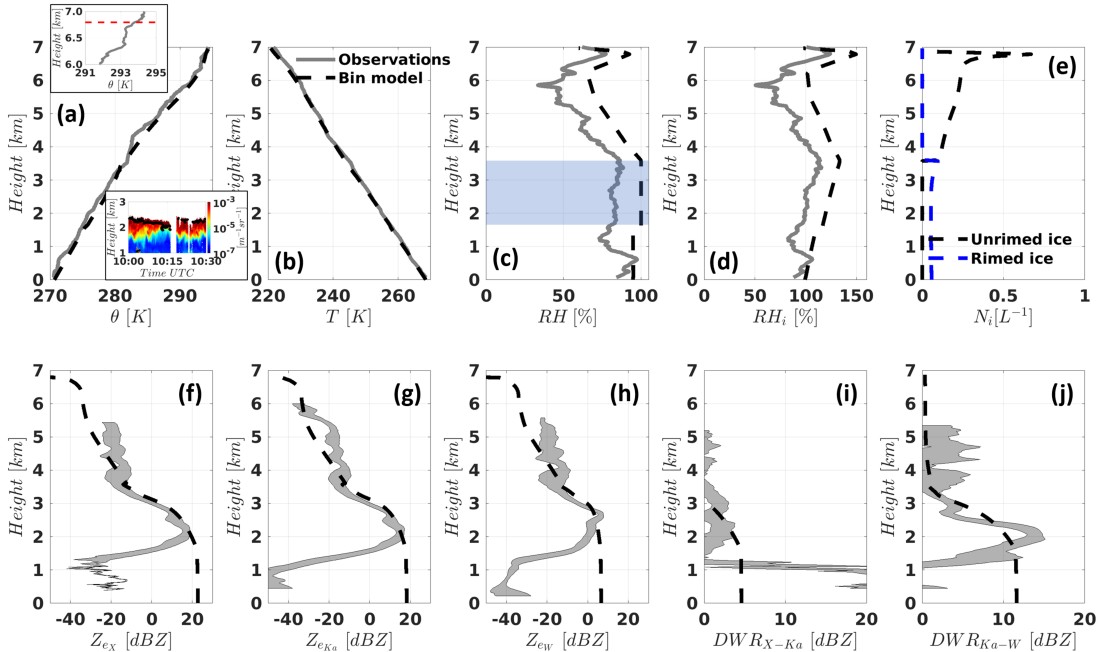

**Figure 13.** Sounding (10:24 UTC) and radar (09:30-09:55 UTC) measurement profiles together with the bin model output at the end of the 8-h simulation. (a) Potential temperature ($\theta$), (b) temperature, (c) relative humidity (RH), (d) RH with respect to ice ($RH_i$), (e) unrimed and rimed ice number concentration (model only; see legend), (f) measured temporal mean $\pm 1$ standard deviation XSACR (shaded region) plus model forward-calculated X-band (black dashed line) reflectivities, (g) as in panel f but for KAZR (Ka-band), (h) as in panel f but for MWACR (W-band), (i) as in panel f but showing the $DWR_{X,Ka}$, and (j) as in panel f but showing the $DWR_{Ka,W}$. The blue-shaded rectangle in panel c represents the estimated supercooled cloud extent (see text). The insets in panel a depict the $\theta$ profile between 6 and 7 km ASL altitude with the dashed red line designating the RH peak (see panel c) at 6.8 km ASL (top), and backscatter cross section measured with the HSRL around radiosonde release time together with liquid cloud base heights (black markers; see Silber et al., 2018b, c).

As indicated by the cloud base height product (Silber et al., 2018b) applied on the high spectral resolution lidar (HSRL; Eloranta, 2005) measurements (black dots in Fig. 4a and Fig. 13a's inset), a distinct supercooled cloud layer was continuously

observed over McMurdo Station at altitudes of 1.7-2.5 km ASL. Nevertheless, the sounding relative humidity (RH) measurements (Fig. 13c) show too low values at these altitudes peaking at 84%. Considering the cloud field extent during the event based on the satellite measurements (figs. 2 and 3), we postulate that even with the radiosonde horizontal drift of a few kilometers away from McMurdo until it reached an altitude of 2.5 km ASL, it could not reach a cloud-free layer. Therefore, we deduce that the radiosonde RH measurements became negatively biased at some point of balloon's flight by up to ∼16%. We note that such negative RH measurement biases were detected in some other cases during AWARE based on HSRL, KAZR, and sounding cross-validation (not shown), and were occasionally occurring at other sites as well. Based on this deduction and supported by indications of a geometrically-thick liquid water layer suggested by a consistent hydrometeor population observed in the KAZR spectra (not shown), we conclude that a deep supercooled layer was extending from about 1.7 to 3.6 km ASL, with a possible break of a few hundred meters centering at ∼2.6 km ASL (Fig. 13c). The location of this deep supercooled layer suggests that even if some vertical mixing did occur during this event (at 3.5-4.2 km ASL and 6.4-6.7 km ASL; see vertical potential temperature profiles in Fig. 13a), it mostly took place at altitudes where the ice particle population did not yet experience rapid mass growth due to ice supersaturation conditions (Fig. 13d) and/or intense riming, thereby hindering PSD broadening. The potentially mixed layer between the surface and 1.2 km ASL was below the DWR signatures around 2 km ASL examined in this study, and hence, had no impact on the observed signatures.

The sounding RH bias also indicates that it is plausible that the ∼100 m thick RH peak value aloft of 77.5% at 6.7 km ASL was actually greater, on the order of 93-94%. The source of this elevated shallow high-RH layer could be associated with the apparently mixed ∼300 m deep underlying layer potentially transporting relatively warmer and moister air aloft (Fig. 13a and its inset). However, a detailed investigation of this moisture source as well as the generating mechanism of the elevated mixed layer, which could be related to gravity wave breaking, for example (e.g., Lane and Sharman, 2006; Podglajen et al., 2017), is beyond the scope of this study.

The inferred RH peak-values (>90%) in this elevated layer suggest that homogeneous freezing of humidified aerosols (hereafter aerosol freezing; e.g., Jensen and Ackerman, 2006; Jensen et al., 2001) might have played a role in the initial formation of at least some of the ice population examined in this study, owing to the large $RH_i$ values (Fig. 13d), which occurred close to water saturation at a low temperature of -50°C measured at the RH-peak altitude (Fig. 13b). Continuous precipitation of ice from an altitude of ∼7 km ASL is indicated by spaceborne radar echoes over the region (Fig. 2) as well as by the ground-based radar echoes (below ∼6 km ASL; Fig. 4). The difference between the topmost radar echoes in the ground-based versus spaceborne measurements could be the result of relatively small ice particles between 6-7 km ASL (especially if indeed formed via aerosol freezing) combined with the limited ground-based radar detectability at the upper troposphere (e.g., Silber et al., 2021).

Following this discussion while acknowledging that our constraining observations over the depth of the atmospheric profiles are relatively limited in space and time, we use a simplified approach, where possible, to initialize and run the bin model. Thus, we run the model over a 1-dimensional (column) domain justified by generally stable atmosphere. We also nudge the bin model simulation thermodynamically (liquid potential temperature and total water) to the local sounding measurements over McMurdo Station (Figs. 2 and 13) using a nudging time scale of 15 min. That is, because the long duration of the triple-frequency signature (∼5 h) suggests steady-state Eulerian conditions. Moreover, the highly-complex flow fields typical to the

McMurdo region (e.g., Silber et al., 2019a, Fig. S1) often result in large reanalysis and regional model biases in reproducing local flow patterns and thermodynamic fields (see Silber et al., 2019c), inhibiting the option of faithfully informing the Eulerian column model with advective tendencies. With the implemented 15-min nudging time scale, our 8-h long sensitivity test simulations typically reach steady state after 3-5 h, allowing us to process and examine the bin model output profiles at the end of the simulation.

Our simplified approach is also incorporated in our treatment of the initialized (and nudged) thermodynamic profile as well as in the treatment of ice nucleation. Ice nucleation is represented only via homogeneous freezing of humidified ammonium bisulfate aerosol or activated droplets (heterogeneous ice nucleation is neglected). Since only aerosol freezing occurs here, the ice number concentrations are effectively determined by the RH maximum within the most elevated moisture layer. In the simulation discussed below, which provided rough agreement with observations, the model is being nudged to an RH peak value of 94%, consistent with the RH bias discussion above. We note that results similar to the examined simulation at the bottom 3.5 km ASL were also obtained in different model simulations (not shown) in which ice nucleation via heterogeneous immersion freezing was included and presented the only source of ice crystals (without aerosol homogeneous freezing aloft, thus omitting the mid-to-upper tropospheric ice); in that case, ice nucleation was concentrated at $\sim$3.6 km ASL, at the top of the deep supercooled layer.

The RH profile below the moisture layer peak is set such that the full profile is supersaturated with respect to ice (Fig. 13d), thereby excluding potential ice sublimation and growth convolution effects on the model output. The RH profile is set to be supersaturated between $\sim$1.7-3.6 km ASL (Fig. 13c), enabling the formation and persistence of the deep supercooled layer in steady-state. The sounding temperature measurement profile is kept unmodified (Fig. 13). Aerosol are set to a log-normal PSD with a mean diameter of 0.076 $\mu m$, a geometric standard deviation of 1.5, and total concentration of 430 $cm^{-3}$, the values of which are based on monthly mean surface measurement at AWARE for January 2016 (Liu et al., 2018).

The bin model is initialized with a single liquid water group and two ice groups: one for unrimed (pristine) ice and another for rimed ice. Each hydrometeor group consists of 60 bins with a minimum radius of 1 $\mu m$ and mass ratio between consecutive bin radii of 1.5, allowing maximum particle diameter of a few centimeters. For the unrimed ice group, we use mass- and area-diameter power law parametrizations for radiating plates taken from Fridlind et al. (2012, Table 1), which generally correspond with the polycrystal ice habit regimes (Bailey and Hallett, 2009) of the generating layer, whether it was actually the elevated moisture layer at $\sim$6.7 km ASL or at the top of the deep supercooled layer at $\sim$3.6 km ASL (Fig. 13). The rimed ice mass- and area-diameter power law parametrizations we use are the SSRGA-derived parameters for $\mathcal{M} = 0.016$ discussed in Sect. 4.4. Collision and accretion between droplets (creating larger droplets or drops), between unrimed ice and droplets (converting to rimed ice), and between rimed ice and droplets (increasing rimed ice mass) are computed from pairwise particle properties (masses, maximum dimensions, aspect ratio, and projected areas) following Böhm (1999, 2004). Aggregation of ice particles is neglected. Consistently, the forward radar calculations are performed using the same SSRGA method informed by the bin model output.

### 4.6.2 Model results

The model simulation reached steady-state conditions after $\sim$5 h. During steady-state, in-cloud mean droplet number concentration is $\sim$30 cm$^{-3}$. The domain's LWP is $\sim$220 g m$^{-2}$, in general agreement with rough LWP estimates on the order of 100-200 g m$^{-2}$ using the method developed by Tridon et al. (2020). Ice water path (IWP) retrievals following Hogan et al. (2006) using the sounding temperature and KAZR reflectivity measurements suggest values during the event on the order of 400 g m$^{-2}$ (see Fig. 11), the highest IWPs observed during AWARE. The model steady-state IWP values of $\sim$165 g m$^{-2}$ are in general agreement with these retrievals considering the high uncertainty associated with this (and other) radar-based ice water content retrievals (e.g., Heymsfield et al., 2008).

Fig. 13 illustrates profiles of ice particle concentrations (panel e) together with forward calculated reflectivities and DWRs (panels f-j) corresponding to the end of simulation (8 h). Initial ice nucleation occurs at the elevated RH layer via aerosol freezing with a maximum ice number concentration of $\sim$0.7 L$^{-1}$ (Fig. 13e). The ice number concentration apparently decreases with height because the ice particle fall velocities increase with decreasing height as their mass becomes larger due to vapor growth under the ice-supersaturated conditions, reaching roughly uniform concentrations with height that are consistently smaller than retrieved, as discussed further below. As the precipitating ice particles reach an altitude of $\sim$3.6 km ASL, the ice particles become rimed and quickly gain additional mass.

Intensification of the Ka- and X-band reflectivities at these lower altitudes (from 3.5 to 2.1 km ASL) in which the deep supercooled layer exists (Fig. 13f-g; the W-band reflectivity intensification in Fig. 13h is less pronounced) is commensurate with this rapid mass growth. These reflectivity strengthening patterns and values down to the reflectivity peak (between 2.1-3.5 km ASL) are in reasonable agreement with the radar observations. Note that reflectivity aloft is underestimated in this model simulation by up to several dB, which could be attributable to possible biases in the representation of unrimed polycrystals or underestimated ice number concentrations.

The model-based DWRs within the deep supercooled layer where the triple-frequency signature was detected show key similarities with the observations; that is, the DWR$_{X,Ka}$ is kept at low values ($< 5$ dB) while the DWR$_{Ka,W}$ increases to large values of $\sim$12 dB (Fig. 13i-j). These similarities are also evident from Fig. 14a, which shows the spread in the observed DWR values at different height ranges together with the temporally-averaged values and the bin model output.

To examine the correspondence of this case study's observational analysis conclusions concerning the shape of the ice PSD able to generate the observed triple-frequency signatures (Sect. 4), we perform a gamma distribution fitting to the bin array profile of the rimed ice group at the end of the simulation. Fig. 14b shows a profile of $\mu$ derived from gamma distribution fits that agree reasonably well with the rimed ice PSD (adjusted $r^2 > 0.98$). The $\mu$ profile indicates a very narrow PSD ($\mu > 40$) at the top of the supercooled layer followed by stabilization of $\mu$ values at a range of 9-11 at lower heights, which is consistently lower than the best fit to observations ($\mu \sim 20$). The mean and median ice particle diameters range between 3 and 4 mm over the height range around 2 km ASL corresponding to the DWR signatures (Fig. 14c).

Taken together, these model results offer general support for the observational and theoretical analysis of the triple-frequency signatures detected during this January 4 2016 event (Fig. 9). Although simulated number concentrations are lower than re-

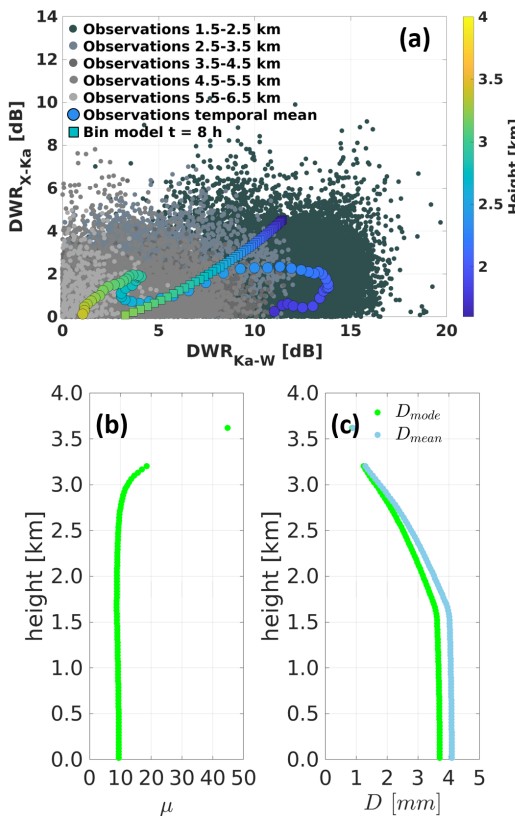

**Figure 14.** (a) DWR parameter space scatter plot showing the observations (09:30-09:55 UTC) at different height ranges, the temporal-mean observations (color scale denotes height), and the bin model output at the end of the simulation (8 h) (see legend). (b) Shape parameter ($\mu$) derived from gamma distribution fits for the rimed ice group at different heights. $\mu$ values are only shown for fits in which the adjusted $r^2$ is larger than 0.98. (c) Mode and mean diameter of the rimed ice group using the gamma fit parameters under the same adjusted $r^2$ threshold.

trieved and the PSD shapes not as narrow, the overall development of DWR trends with height is generated by the model's standard physics schemes without requiring any special tuning. If model number concentrations are doubled prior to the forward calculations, reflectivity is overestimated relative to the observations (not shown). We have therefore not tried to exactly match retrieved number concentrations and ice PSD widths, also owing to weak constraints on ice crystal properties.

In summary, the bin model simulation corroborates the plausibility of scenarios capable of producing the unique signatures observed during this event. DWR parameter space agreement with the observations could be reached in other sensitivity tests where some vertical motion was introduced or in cases where the nudged sounding profile was slightly modified. However, reaching these high (low) $DWR_{Ka,W}$ ($DWR_{X,Ka}$) values always required some compromises concerning the reflectivity aloft, for example (as in this depicted simulation), thus emphasizing the difficulty to constrain ice properties with our limited obser-

vational dataset. Yet, the fact that similar results could be reached using somewhat varying configurations as long they allowed narrow rimed ice PSD with diameters on the order of a few millimeters to be produced and developed, suggest that such cases

might occasionally occur over the Antarctic (or other polar regions). The frequency of occurrence of such scenarios will be examined in future studies.

## 5   Conclusions

In this work, by exploiting the deployment of an unprecedented number of multi-wavelength remote sensing systems (including triple-frequency radar measurements) at McMurdo station, Antarctica, during the Atmospheric Radiation Measurements West Antarctic Radiation Experiment (AWARE) field campaign, we find frequent occurrences of high Ka-W dual-wavelength ratios (DWR) coinciding with relatively low X-Ka dual-wavelength ratios taking place at unexpectedly low temperatures of -20°C, in comparison to the mid-latitudes. These features, generally interpreted as riming signatures, suggest a likely common atmospheric state over Antarctica that includes a rather stable atmosphere inhibiting turbulent mixing, and a high riming efficiency driven by large cloud droplets. We note, however, that the limited duration of the triple frequency dataset collected during AWARE does not allow drawing definite conclusions concerning the frequency of such events.

A peculiar case study is analysed in greater detail: it features a persistent layer with relatively modest amounts of supercooled liquid water producing particularly strong riming signatures in triple-frequency radar data. Since in-situ observations are lacking, the radar observations are exploited to retrieve the properties of the ice particles leading to these signatures. To this aim, several state-of-the-art microphysics and scattering models (EM-MIC models) are used. The combination of the triple-frequency radar reflectivities, the differential spectral width and a proxy of the ice particles fall velocities derived from the Doppler spectra allows us to constrain the microphysical properties of the ice particles. Results suggest that the fall velocity associated with recent rimed aggregates EM-MIC models is too large, and a novel potentially more realistic EM-MIC model is therefore proposed. Even if a non-negligible uncertainty remains on the size of the retrieved ice particles, results indicate that the observed DWR signatures can only be explained by the combined effects of moderately rimed aggregates or similarly shaped florid polycrystals and a narrow particle size distribution (PSD). More studies are needed to validate the retrieval algorithm proposed here. This could be done either by cross-comparing the algorithm results with other techniques and/or by using in-situ validation datasets from field campaigns (Leinonen et al., 2018; Mason et al., 2018; Battaglia et al., 2020b; Mroz et al., 2021; Nguyen et al., 2021), while noting that commonly-used airborne imagers are limited to particles sizes smaller than those deduced in this study.

Simulations of this case study performed with a 1D bin model confirm that, with the modest amount of supercooled liquid water, the triple frequency radar observations can be generally reproduced, provided that narrow PSDs are simulated. Such narrow PSDs could be explained by two key factors: (i) the presence of a shallow homogeneous droplet or humidified aerosol freezing layer aloft seeding a supercooled liquid layer, and (ii) the absence of turbulent mixing throughout a stable polar atmosphere that sustains narrow PSDs, as hydrometeors grow from the nucleation region aloft to several millimeter ice particles, by vapor deposition and then riming.

This study illustrates that triple-frequency radar measurements can be used to infer detailed properties of precipitating ice such as the PSD width or the degree of riming of ice particles. While the associated retrieval techniques are still at an exploratory

stage, such information is crucial for improving our understanding of the role of ice phase in the water budget. Therefore, more observations and analysis involving triple-frequency radars are needed in the future.

*Data availability.* AWARE data were obtained from the U.S. DOE ARM Climate Research Facility https://www.archive.arm. gov (Atmospheric Radiation Measurement (ARM) user facility, 2014, 2015a, c, b).

*Author contributions.* ARM radars data analysis was made by FT and PK. Bin model simulations and their analysis were made by IS and 670 AF. Interpretation and writing were shared between FT, IS, AB, SK and AF. RD provided the satellite data.

*Competing interests.* The authors declare that they have no conflict of interest.

*Acknowledgements.* This work was funded by the US Atmospheric System Research (grant no. DESC0017967). Contributions by SK were funded by the German Research Foundation (DFG) under grant KN 1112/2-1 as part of the Emmy-Noether Group "Optimal combination of Polarimetric and Triple Frequency radar techniques for Improving Microphysical process understanding of cold clouds" (OPTIMIce). We 675 thank the two anonymous reviewers for their detailed and thorough comments which helped greatly to improve the manuscript.

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
