# Peer review of "Highly supercooled riming and unusual triple-frequency radar signatures over McMurdo station, Antarctica"

_Atmospheric Chemistry and Physics, 2022_

## Referee Comment (RC1)

**General comments:**

In this manuscript, the authors quantitatively interpret triple-frequency radar signatures in snow and ice clouds observed in Antarctica. Their results indicate that riming starts at lower temperatures in Antarctica than at mid-latitudes and that observed triple-frequency radar signatures with extreme values of DWR_Ka,W (dual-wavelength ratio of Ka and W band reflectivities) can only be explained by approximating the particle size distribution by gamma distributions with high shape parameters mu (called a 'narrow' particle size distribution).

Overall, the manuscript is well written and presents the main points clearly. Nonetheless, some parts of the manuscript could still benefit from more careful editing to improve the grammar and the clarity of the arguments made (the 'Technical corrections' suggested below can provide a starting point).

I think, while some of the points discussed in the manuscript (lengthy discussion of Fig. 4, entire section 4.6, see specific comments below) may add value to the study, they are also difficult to follow and therefore dilute the main results. One important point that could still be discussed in a revised version is how realistic the narrow particle size distributions are that form the basis for the retrieval of ice properties from the triple-frequency radar signatures (see also comment regarding Section 4.4, 4.5 below), given the somewhat extreme values of the PSD shape parameter and the missing information about other commonly used parameters of the size distribution(s).

Considering the high quality of the study and the novelty of the results, I would therefore suggest to publish the manuscript after these points are addressed.

**Specific comments:**

**l. 137 ff**: Can the authors quantify these disparities, because these disparities seem to form the motivation for the entire analysis?

For example, a very basic method would be to compare (i) by how much mean or median DWR_Ka,W increases from -25 °C to -15 °C for both locations (also in relation to the width of the distributions with width interpreted e.g. as range from 10th to 90th percentile or standard deviation) and (ii) how much of the DWR_Ka,W distributions is 'much higher than the common maximum of 12 dB'. The general goal here should be to provide some reasonable parameter(s) for how large the disparities are and how (statistically) significant they are for the given datasets. No need for a detailed statistical analysis.

**l. 144 ff**: Could this low aerosol concentration or a different type of aerosol found over Antarctica not also cause the initiation of aggregation at a lower temperature? What about characteristic differences in the typical wind field, could those affect aggregation/riming temperatures and lead to the initiation of aggregation at a lower temperature in Antarctica? Maybe the authors can present a few (more) arguments to support their conclusion that the differences in observed triple-frequency signatures can only be attributed to the riming process.

**Fig. 4 and section 4.2**: Sooooo many plots in a single figure. In my opinion, the discussion of this figure also follows many trains of thought that probably all make sense but I could not grasp all of them. To appropriately make all points that the authors intend to make here, more (con)text and multiple separate figures would be needed. I would therefore suggest to focus only on the most relevant points of this entire discussion and omit the rest.

**l. 228 ff**: I cannot follow this conclusion. Can the authors elaborate a bit or rephrase to clarify?

**Section 4.4, 4.5**: What range of Lambda values is used for the calculations to obtain the lookup table for the retrievals? Do all the retrieved parameters of the size distributions and the overall ranges used for the retrievals represent realistic size distributions in snow and ice clouds?

For example, Brandes et al. (2007, DOI: 10.1175/JAM2489.1) rarely observed Gamma-PSD shape parameters of mu > 10 in their in situ observations at mid-latitudes. Gergely (2019, DOI: 10.1016/j.jqsrt.2019.106605) found a strong impact of the maximum snowflake diameter of the chosen particle size distribution on radar retrievals. Other studies also suggest that the slope parameter Lambda spans only a somewhat limited range of values. How do the values that the authors use and the ones they ultimately retrieve compare to those discussed in other studies? Are they similar? Or is there any reason why a different range of values may be appropriate to describe particle size distributions in Antarctica?

Maybe the authors could also plot some of the size distributions that they retrieve or just some generic Gamma-PSDs to illustrate what a 'narrow' size distribution looks like vs. a 'wide' size distribution, as understood by the authors in the context of this study (confusion can arise here because the parameter Lambda can also be interpreted as a measure of the width of the distribution).

**l. 350 and following paragraphs**: Substitute 'measurement uncertainties' and 'retrieval uncertainties', etc. for 'errors'. These are uncertainties, not errors.

**Section 4.6**: While this section describes an interesting exercise, it is not entirely clear to me whether this chapter adds anything substantial to this study, particularly because the results are mostly qualitative and can be interpreted as a type of consistency check. In my opinion, this does not add a lot of signifant results to the results presented in the previous chapter(s). I would instead (or additionally) prefer to see a brief discussion on how realistic the 'narrow' particle size distributions are, e.g., based on a comparison/discussion of studies that have used and obtained relevant parameters of snow and ice particle size distributions from in situ observations (see also comment regarding Section 4.4, 4.5).

**Technical corrections:**

l. 69: 'data' instead of 'Data'?

l. 144: 'this difference' or 'these differences', but not 'this differences'

Fig. 2 caption: 'temperature profiles'

l. 188: Do you mean ... 'refractive index' ...

l. 193: I do not understand ' where the temperature is comprised between -25 and -40_C', please rephrase, maybe you mean 'where the temperature is between -25 and -40 °C'.

l. 212 ff: Can you rewrite this discussion, so it is more easily understood. For example, I get confused by multiple clauses starting with 'conversely' so close to each other.

l. 214: Do you mean ... vertical 'bands' ... that alternate between blue and red?

l. 224: spectrum width (singular)

l. 226: Well, apparently they are not identical. Why not use 'very similar' or something along those lines?

Fig. 7 caption: replace 'comprised between' with 'of'

l. 388: delete 'notable'

l. 394: Likewise, Fig. 9 c ... (instead of 'Likewise, the figure 9 c')

l. 421: What does 'liquid-free layer' mean? Without liquid water present?

l. 424: What is a 'geometrically-thick liquid water hydrometeor population'. Can you rephrase this to make it clearer?

l. 508: maybe better to write something like 'These similarities are also evident from Fig. 11a which shows the spread ...'

l. 545: ... allows us to constrain the microphysical properties of the ice particles ...

---

## Author Comment (AC1)

**acp-2022-136**

**Responses to reviewer 1**

Highly supercooled riming and unusual triple-frequency radar signatures over Antarctica

F. Tridon, I. Silber, A. Battaglia, S. Kneifel, A. Fridlind, P. Kalogeras, and R. Dhillon

June 7th, 2020

We thank the reviewer for his efforts and time for reviewing as well as constructive comments which greatly helped to improve the manuscript. All our point-to-point answers are highlighted in red below according to the following sequence: (i) comments from referees/public, (ii) author's response, and (iii) author's changes in manuscript.

**General comments:**

In this manuscript, the authors quantitatively interpret triple-frequency radar signatures in snow and ice clouds observed in Antarctica. Their results indicate that riming starts at lower temperatures in Antarctica than at mid-latitudes and that observed triple-frequency radar signatures with extreme values of DWR Ka,W (dual-wavelength ratio of Ka and W band reflectivities) can only be explained by approximating the particle size distribution by gamma distributions with high shape parameters mu (called a 'narrow' particle size distribution). Overall, the manuscript is well written and presents the main points clearly. Nonetheless, some parts of the manuscript could still benefit from more careful editing to improve the grammar and the clarity of the arguments made (the 'Technical corrections' suggested below can provide a starting point). I think, while some of the points discussed in the manuscript (lengthy discussion of Fig. 4, entire section 4.6, see specific comments below) may add value to the study, they are also difficult to follow and therefore dilute the main results. One important point that could still be discussed in a revised version is how realistic the narrow particle size distributions are that form the basis for the retrieval of ice properties from the triple-frequency radar signatures (see also comment regarding Section 4.4, 4.5 below), given the somewhat extreme values of the PSD shape parameter and the missing information about other commonly used parameters of the size distribution(s). Considering the high quality of the study and the novelty of the results, I would therefore suggest to publish the manuscript after these points are addressed.

**Specific comments:**

1) I. 137 ff: (i) Can the authors quantify these disparities, because these disparities seem to form the motivation for the entire analysis? For example, a very basic method would be to compare (i) by how much mean or median DWR\_Ka,W increases from -25 °C to -15 °C for both locations (also in relation to the width of the distributions with width interpreted e.g. as range from 10th to 90th percentile or standard deviation) and (ii) how much of the DWR\_Ka,W distributions is 'much higher than the common maximum of 12 dB'. The general goal here should be to provide some reasonable parameter(s) for how large the disparities are and how (statistically) significant they are for the given datasets. No need for a detailed statistical analysis.

(ii) (iii) We added this statistical information and modified the text and figures.

- 2) I. 144 ff: (i) Could this low aerosol concentration or a different type of aerosol found over Antarctica not also cause the initiation of aggregation at a lower temperature? What about characteristic differences in the typical wind field, could those affect aggregation/riming temperatures and lead to the initiation of aggregation at a lower temperature in Antarctica? Maybe the authors can present a few (more) arguments to support their conclusion that the differences in observed triple-frequency signatures can only be attributed to the riming process. (ii) The main argument supporting that the differences are not due to the aggregation process are the triple-frequency signatures themselves: as recalled in section 3.1, aggregation would cause an increase in DWRx.Ka practically equal to the increase in  $DWR_{Ka,W}$  while no increase in the AWARE  $DWR_{X,Ka}$  is seen above the -15°C level (i.e., level from which aggregation is expected to be efficient). We are therefore confident that the differences in the triple-frequency signatures are due to riming. What is less certain is the reason why riming is happening at a lower temperature in the AWARE dataset. We agree that we cannot exclude the potential effect of wind due to the complex topography around McMurdo station. (iii) We have added this hypothesis and we modified the text in order to better explain that the uncertainty is on which mechanism leads to riming at lower temperature but not on the occurrence of riming.
- 3) Fig. 4 and section 4.2: (i) Sooooo many plots in a single figure. In my opinion, the discussion of this figure also follows many trains of thought that probably all make sense but I could not grasp all of them. To appropriately make all points that the authors intend to make here, more (con)text and multiple separate figures would be needed. I would therefore suggest to focus only on the most

relevant points of this entire discussion and omit the rest.

(ii) (iii) In our opinion, all the points discussed are important. In order to make them more understandable, we separated the Fig. 4 in 3 different figures of 3 panels (i.e., we were able to remove one panel), and we describe them in 3 different subsections (4.2.1, 4.2.2 and 4.2.3), and we reworded some of the text.

4) **I. 228 ff:** (i) I cannot follow this conclusion. Can the authors elaborate a bit or rephrase to clarify?

(ii) (iii) We rephrased this paragraph (now first paragraph of section 4.2.3) to make it clearer.

5) Section 4.4, 4.5: (i) What range of Lambda values is used for the calculations to obtain the lookup table for the retrievals? Do all the retrieved parameters of the size distributions and the overall ranges used for the retrievals represent realistic size distributions in snow and ice clouds? For example, Brandes et al. (2007, DOI: 10.1175/JAM2489.1) rarely observed Gamma-PSD shape parameters of mu > 10 in their in situ observations at mid-latitudes. Gergely (2019, DOI: 10.1016/j.jgsrt.2019.106605) found a strong impact of the maximum snowflake diameter of the chosen particle size distribution on radar retrievals. Other studies also suggest that the slope parameter Lambda spans only a somewhat limited range of values. How do the values that the authors use and the ones they ultimately retrieve compare to those discussed in other studies? Are they similar? Or is there any reason why a different range of values may be appropriate to describe particle size distributions in Antarctica? Maybe the authors could also plot some of the size distributions that they retrieve or just some generic Gamma-PSDs to illustrate what a 'narrow' size distribution looks like vs. a 'wide' size distribution, as understood by the authors in the context of this study (confusion can arise here because the parameter Lambda can also be interpreted as a measure of the width of the distribution).

(ii) One of the core result of our study is that the retrieved PSDs are indeed unusually narrow, and hence our retrieved shape parameter mu values are in the upper part of its usual range. Nevertheless, other parameters such as Dm, concentration and slope parameter Lambda lie in the range commonly found in literature.

(iii) Because we think that there are some redundant information between Dm and Lambda, and to avoid a too large number of figures in the manuscript, we only show the Lambda values retrieved with the SSRGA-plates-M0.016 model in the Figure below. Nevertheless, we added a new figure in the manuscript (now Figure 12) in order to show examples of the gamma PSDs retrieved and their corresponding forward-modeled Doppler spectra. The figure demonstrates that the retrieved PSDs are compatible with the observed Doppler spectra. We now discuss these issues in several parts of section 4.5.

- 6) I. 350 and following paragraphs: (i) Substitute 'measurement uncertainties' and 'retrieval uncertainties', etc. for 'errors'. These are uncertainties, not errors.
  (i) We thank the reviewer for noticing this! (iii) Corrected
- 7) Section 4.6: (i) While this section describes an interesting exercise, it is not entirely clear to me whether this chapter adds anything substantial to this study, particularly because the results are mostly qualitative and can be interpreted as a type of consistency check. In my opinion, this does not add a lot of signifant results to the results presented in the previous chapter(s). I would instead (or additionally) prefer to see a brief discussion on how realistic the 'narrow' particle size distributions are, e.g., based on a comparison/discussion of studies that have used and obtained relevant parameters of snow and ice particle size distributions from in situ observations (see also comment regarding Section 4.4, 4.5).

(ii) We respectfully disagree with the reviewer. We think that Section 4.6 is essential to this study, mostly because it addresses the reviewer's main concerns in this comment and in the general comment regarding how realistic our concluded PSD is. Comparisons to the literature are not relevant in this case for the following reasons:

- 1. In-situ airborne observational datasets of Antarctic ice PSDs are scarce and contain most samples at temperatures greater than -15.
- 2. The airborne instruments commonly used to obtain these ice PSDs (e.g., CPI, CIP) are limited to maximum dimensions on the order of 1.5 microns, smaller than the sizes derived here of a few to several microns, making studies using these instruments irrelevant.
- 3. Lower-latitude clouds exhibit different characteristics (number concentrations, etc.) than Antarctic clouds so retrieved ice PSDs from such lower-latitude studies are irrelevant.
- 4. Antarctic ground-based measurements and retrievals are confounded by commonly-occurring low-level ice sublimation (driven by Foehn winds over the

region) and blowing snow, and therefore, are challenging to objectively compared with our results.

Instead, we wish to provide physical context to our retrieval and results with this modeling exercise, in which we demonstrate that such narrow PSDs are realistic, and that the simulated scenario based on the full AWARE instrument suite measurement serves as a plausible explanation for the observed signatures.

(iii) We now explain that in the first paragraph of sect. 4.6:

"In-situ airborne observations of ice PSDs over Antarctica are relatively scarce and are commonly performed using instruments such as the Cloud Particle Imager (CPI Lawson et al., 2001) and the Cloud Imaging Probe (CIP Lachlan-Cope et al., 2016). which are limited to a particle maximum dimension of ~1.5 mm. This lack of a comprehensive observational database of Antarctic ice precipitation PSDs combined with instrument-detectable particle size limitations inhibits any pertinent comparison with the results presented here of ice particles with sizes on the order of a few to several mm generating the observed triple-frequency radar signatures. Moreover, the narrow and elevated altitude range depicting those radar signatures further impedes comparisons to ice particle properties derived from Antarctic ground-based observations, which are strongly influenced by low-level ice sublimation (e.g., Grazioli et al., 2017b) and blowing snow events (e.g., Loeb and Kennedy, 2021), frequently occurring over the region. Instead, we examine using the following modeling exercise whether a plausible riming scenario based on AWARE observations could develop the detected triple-frequency signature, thereby adding a physical context to this analysis, which exemplifies that such narrow ice PSDs are realistic."

We also refer to the instrument-detectable maximum ice size in the conclusions:

"More studies are needed to validate the retrieval algorithm proposed here. This could be done ... by using in-situ validation datasets from field campaigns ..., while noting that commonly-used airborne imagers are limited to particles sizes smaller than those deduced in this study."

Finally, we also modified the section title accordingly: *"4.6: Bin model experiment: can a plausible scenario reproduce the observed narrow PSD of rimed ice particles?"*

**Technical corrections:**

- I. 69: 'data' instead of 'Data'? Done
- I. 144: 'this difference' or 'these differences', but not 'this differences' Done
- Fig. 2 caption: 'temperature profiles' Done
- I. 188: Do you mean ... 'refractive index' ... Yes! Corrected

- I. 193: I do not understand ' where the temperature is comprised between -25 and -40\_C', please rephrase, maybe you mean 'where the temperature is between -25 and -40 °C'. Yes. Corrected
- I. 212 ff: Can you rewrite this discussion, so it is more easily understood. For example, I get confused by multiple clauses starting with 'conversely' so close to each other. Done
- I. 214: Do you mean ... vertical 'bands' ... that alternate between blue and red? Yes. Corrected
- I. 224: spectrum width (singular) Corrected
- I. 226: Well, apparently they are not identical. Why not use 'very similar' or something along those lines? Done
- Fig. 7 caption: replace 'comprised between' with 'of' Done
- I. 388: delete 'notable' Done
- I. 394: Likewise, Fig. 9 c ... (instead of 'Likewise, the figure 9 c') Corrected
- I. 421: What does 'liquid-free layer' mean? Without liquid water present? Yes, changed to "cloud-free".
- I. 424: What is a 'geometrically-thick liquid water hydrometeor population'. Can you rephrase this to make it clearer? Sentence reworded: "Based on this deduction and supported by indications of a geometrically-thick liquid water layer suggested by a consistent hydrometeor population observed in the KAZR spectra (not shown), ..."
- I. 508: maybe better to write something like 'These similarities are also evident from Fig. 11a which shows the spread ...' Text modified per the reviewer's suggestion.
- I. 545: ... allows us to constrain the microphysical properties of the ice particles ... Corrected

---

## Author Comment (AC2)

**acp-2022-136**

**Responses to reviewer 2**

Highly supercooled riming and unusual triple-frequency radar signatures over Antarctica

F. Tridon, I. Silber, A. Battaglia, S. Kneifel, A. Fridlind, P. Kalogeras, and R. Dhillon

June 7th, 2020

We thank the reviewer for his efforts and time for reviewing the manuscript. All our point-to-point answers are highlighted in red below according to the following sequence: (i) comments from referees/public, (ii) author's response, and (iii) author's changes in manuscript.

**General comments:**

The manuscript well explained the techniques of the retrievals used in this study and entire methods of the measurements and retrievals and also evaluated the techniques well including uncertainties owing to observational limitations and assumptions used in the techniques. I really appreciate those descriptions; this helped to interpret the observed features. Figures are all beautiful at high resolution. I had expected more analyses and/or discussions about physics and characteristics of Antarctica, such as mesoscale, microphysical and dynamical processes, as the manuscript had been submitted to ACP (not AMT).

We agree that the title was a bit misleading because it was suggesting that we found a feature specific to the whole Antarctic continent. This is not the case and we cannot generalize our findings and verify their occurrence at other sites in Antarctica due to the lack of equivalent measurements. Therefore, we decided to change the title so that it describes the study more accurately, i.e. that we found a peculiar process at McMurdo station. Describing the mesoscale and dynamical processes of Antarctica (comment 1 below) is out of the scope of the study because it would require additional tools such as meso-scale modeling. Nevertheless, we think that ACP is the right journal because the aim of the paper is not to describe the technical details of a retrieval methodology but a previously undocumented physical process which we investigate both via observations and modeling.

**Specific comments:**

1) I expected a little bit more discussions and/or analyses for the following points at least:

1.1) The analysis results from the Antarctica data were compared with data from only a site. Is this enough to discuss the characteristics of the Antarctic microphysics? Can you take into account other environments such as continental, maritime, coastal, arctic, mountain, etc.?

(i) Such a request cannot be satisfied largely because of the lack of observations. Less than a handful of field campaigns with long term triple-frequency radar measurements have been conducted in the past (the main 3 being the ones mentioned in the paper: AWARE, BAECC and TRIPEX). This also limits our ability to examine the representativeness of our findings to the Antarctic continent. (iii) Therefore, we decided to change the title, abstract, conclusion and description of the results to reflect that we have indeed found a previously undocumented process at McMurdo station.

1.2) What are the environmental characteristics of the site in terms of temperature (like lines 536-538), humidity, wind, vertical velocity, etc.?

(ii) The general description of the environmental characteristics at McMurdo station during AWARE has already been the object of several papers. (iii) We have added the new section 2.1 to summarize these findings and we completed the Table 1 with mean values of temperature, relative humidity and horizontal wind for each case.

1.3) Can the triply-frequency signatures be expected to be generalized for other sizes in Antarctica?

(ii) We guessed that "sizes" was a typo and interpreted it as "sites". It is conceivable that riming could occur at lower temperatures over the whole Antarctic continent due to the low aerosol concentration but this must be corroborated by observations at other sites. (iii) Like for comment 1.1, we no longer argue that this feature is present over the whole Antarctic continent.

1.4) How can the fewer but larger supercooled liquid droplets (lines 144-146) contribute to riming? I guessed that fewer droplets could restrain riming since the chance of collision and accretion could be reduced.

(ii) The increase of the size of the droplets can dominate the effect of their reduction in concentration because collision efficiency strongly increases when going from small cloud droplets ( $10\mu$ m) to slightly larger drizzle ( $30-40\mu$ m). As a result, the chance of collision and accretion depends mostly on the relative fall velocities between the cloud droplet and ice crystal, and on several aerodynamic effects (Pruppacher and Klett, 1997, Wang and Ji, 2000). (iii) We completed the manuscript with this explanation.

**References:**

Pruppacher, H. and Klett, J.: Microphysics of Clouds and Precipitation, Atmospheric and Oceanographic Sciences Library, Springer Nether-lands, 1996.

Wang, P. K., & Ji, W. (2000). Collision Efficiencies of Ice Crystals at Low-Intermediate Reynolds Numbers Colliding with Supercooled Cloud Droplets: A Numerical Study, *Journal of the Atmospheric Sciences*, 57(8), 1001-1009

1.5) Seasonal variability (I know that the data were limited, though...).

(ii) Indeed, it is impossible to assess the seasonal variability of riming occurrence with only 3 months of data. Since the radars are zenith pointing and provide observations over a single location, several years of continuous observations would be necessary to provide a meaningful seasonal variability.

1.6) Line 528-530: Why the narrow rime ice PSD is characterized over polar regions?

(ii) As demonstrated in the model exercise, the narrow ice PSD generally requires a stable atmosphere and large droplets (which increase the riming efficiency). (iii) We modified the text accordingly:

"These features are often interpreted as riming signatures and pinpoint a relatively common atmospheric state over Antarctica that includes a rather stable atmosphere inhibiting turbulent mixing and high riming efficiency driven by large cloud droplets. We note, however, that the limited amount of the triple frequency dataset collected during AWARE does not allow drawing definite conclusions concerning the frequency of such events."

1.7) Lines 536-538: Can you explain a little bit more why the DWR signatures can be characterized as a unique signature over Antarctica?

(ii) They are unique in the sense that they have not been observed in the previous triple-frequency datasets, which, even if those datasets are not numerous, cover a much longer duration than AWARE. (iii) It would not be reasonable to add the technical description of section 3.2 in the conclusion so instead, we added a line in section 3.2 emphasizing the fact that such features are unique. However, we admit that unique is probably too strong and we now use peculiar instead.

2) Please provide a bit more explanation about BAECC; location, period, radar frequencies, case descriptions... This should help to characterize the observed features in this study.
(ii) (iii) We added a paragraph rapidly describing BAECC and providing

references with more detailed descriptions of case studies.

3) This manuscript analyzed one 'extraordinary' case study. Does this represent the other cases listed in Table 1? Need descriptions of the generality of the results from the detailed analysis among the selected cases.

(ii) The case study features riming signatures like the majority of the other

AWARE cases (i.e., increase in DWRKa,W while the AWARE DWRX,Ka remains close to zero for data above the -15°C level), but at an extreme level (as written at the end of section 3.2, the case study has been selected because it features the strongest DWRKa,W). The results from section 3 and 4-5 could be considered as two independent studies (general occurrence of riming during the whole AWARE field campaign vs. study of the mechanisms leading to extreme riming signatures for a single case study) and this is why they are the object of two different paragraphs in the conclusion and summary. (iii) We think that this was already clear in the manuscript so we only slightly modified the transition between those two paragraphs in both the summary and the conclusion, and we better clarified what distinguishes this case study rather than referring to it generally as 'extraordinary'.

- 4) Line 215-216: This sentence did not make sense to me. Maybe a few more explanations would be needed.
   (iii) We expanded the text to make this discussion clearer.
- 5) What could bring the situation of slightly pointing off-zenith? (ii) Since we already explained in the original manuscrit what is the consequence of slightly pointing off-zenith in section 4.2 ("As a result, a small component of the horizontal winds is found along the pointing direction of the mis-pointing radar which explains the observed dVDKa,W difference"), we interpreted the question of the reviewer as what could be its cause.

Note that we are talking of very subtle mispointing (of the order of 1°). The only reason we are able to detect them is thanks to the high sensitivity and narrow beam widths of cloud radars. Even if it is done with great caution, it is a challenge to align the beams of the radars to such a precise direction. During BAECC, it was found that the mis-alignment of the radar beams was eventually due to the thawing of the soil where the container of one of the radars was installed. (iii) We think that this technical information diverts too much from the main results, so we did not include it in the manuscript.

6) Figure 6 and Figure 7 (now figures 8 and 9): The data points were distributed to a large range. I wondered how such grid points with low density data are significant.

(ii) Note that the values indicated are the absolute numbers ( (iii) we modified the title of the colorbars from "n" to "counts" to make it clear). So, the bluish pixels correspond to less than 10 occurrences and they are indeed not significant, this is why we only try to match the yellow to reddish pixels. We could restrain the color scale to occurences larger than 100, but we wanted to show the whole data points in the selected area.

6.1) Please provide the total number of sample size.

(ii) (iii) We added the total number of points (about 130000) in the introduction of Fig. 8 (section 4.3).

6.2) Because of the noisiness, it seems to me that any lines from the particle models cannot represent the observation data for any plots.

(ii) It is true that multi-frequency radar observations are inevitably noisy not only because of the eventual radar volume mismatch, but also because of the intrinsic noisiness of radar measurements. But the main variability here is the natural variability due to the variety of microphysic processes occurring, the complexity of ice particles and the variability of ice PSD shapes (cf following comment 6.3). Therefore, the objective here is to find the scattering model which best match the observations in average.

(iii) We added a new paragraph in section 4.3 to better explain the methodology. We think that it is relatively clear that some theoretical lines match better the different density plots but to make the comparison more evident, we added the median and  $10^{th}$  and  $90^{th}$  percentiles of the density plot as function of DWRKa,W (white lines in Fig. 8 and 9) to highlight the average trend and natural variability of the observations which can then be compared to the theoretical model lines.

Also, we have now estimated the noisiness of radar measurements for the configuration of the ARM radars during AWARE and we have added error bars in the top-right corner of Fig. 8 and 9 to illustrate them).

6.3) I supposed that the data to plot those figures came from the long time period (~4 hours), which period possibly included a variety of microphysical processes; not only riming, but also depositional growth, different degree of riming, aggregation, etc.. Those data could be plotted into a panel. So I was not sure of a meaningful of those plots; what is the purpose of overlaying the lines from the particle models; why the only selected particle types from models were plotted.

(ii) As it was indicated in the introduction of Fig. 8 (section 4.3), the histograms correspond indeed to the 9h to 13h period. (iii) We modified the text to make it more evident.

(ii) Indeed, the microphysic properties and processes involved during the 4 hours period are various. As explained in the reply to comment 6.2, the objective is to find the scattering model which best match the observations in average, and hence to detect, from the observed reflectivity signatures, the fingerprints of the dominant microphysical process in shaping the ice particles. Note that the main difference between these SSRGA-LS15-Bxxx models is the quantity of supercooled water accreted to the ice particle, i.e. the degree of riming, but during the generation of the ice aggregates, depositional growth and aggregation are intrinsically involved by the explicit simulation of the aggregation of monomers of various sizes. (iii) We added these explanations in section 4.3.

6.4) Because of those, it was unclear for me what is the 'unusual' triple-frequency signatures. It would help if the signatures were highlighted in

the plots.

(ii) (iii) We added a magenta ellipse in figure 8 to highlight the unusual triple-frequency signatures and we expanded its description.

6.5) Lines 372-373 "At the top of the layer..." This sentence does not make sense to me. Please provide more explanation.

(ii) (iii) We replaced the sentence by "From 2.6 to 2.3 km, Dm strongly increases towards the ground, highlighting the layer where riming is most efficient and the probable top of the supercooled liquid layer".

7) Line 428-429: Please explain this process more in details.

(ii) (iii) We modified the text:

"The location of this deep supercooled layer suggests that even if some vertical mixing did occur during this event, ... it mostly took place at altitudes where the ice particle population did not yet experience rapid mass growth due to ice supersaturation conditions... and/or intense riming, thereby hindering PSD broadening."

8) Figure 10:

8.1) Please highlight the location of the supercooled liquid layer(ii) (iii) We added a blue-shaded rectangle to panel c representing the estimated supercooled cloud extent.

8.2) Add a plot and discussion of vertical velocity

(ii) The mean Doppler velocity signatures during the event are already discussed in sect. 4.2.2 and presented in Fig. 5, and since we do not think that another panel to the 10-panel Fig. 10 would add additional insights to the model results discussion, we prefer to leave the figure as is.

**8.3) Why Z\_Ka**

Reference:

Silber, I., Verlinde, J., Eloranta, E. W., Flynn, C. J., and Flynn, D. M.: Polar Liquid Cloud Base Detection

Algorithms for High Spectral Resolution or Micropulse Lidar Data, Journal of Geophysical Research: Atmospheres, 123, 4310–4322, https://doi.org/https://doi.org/10.1029/2017JD027840, 2018.

 Sometimes it was unclear for me that which Ka-band radar (KaSACR or KAZR) was used to estimate DWR\_X/Ka and DWR\_Ka/W? I suppose that KAZR was used throughout the study; why wasn't Ka-SACR used? I expect that the use of Ka-SACR can reduce the beam mismatching error at least for DWR\_X/Ka.

(ii) As described in the Radar data processing section (section 2) and as indicated by the dataset citations, the KaSACR data is not used in this study. The main reason is the better sensitivity of the moderate sensitivity mode (MD) of the KAZR. The use of KaSACR data would indeed avoid the beam mismatch issue, but only where KaSACR SNR is high enough. Instead, we avoided the mismatch issue by using the differential spectral width and not the differential fall velocity. (iii) The choice of KAZR for its sensitivity was indicated in section 2.

- Table 1: Add a temperature range for each case.
   (ii) (iii) Done
- Just I was surprised that WACR had a huge offset in reflectivity (19 dBZ)... Was the sensitivity of the radar enough?

(ii) We were also surprised by the huge offset of the WACR. The resulting sensitivity of the WACR was indeed lower than the sensitivity of the KAZR (as can be deduced by comparing the extent of the KAZR reflectivity field (Figure 4a) with the DWRKaW field (Figure 4e)). This would be a problem if looking at thin ice clouds, but this does not prevent our analysis because we focus on thick clouds generating large ice crystals which lead to non-zero DWRs and significant signal to noise ratio (SNR). Furthermore, the large density of points with DWRs=0 in Figure 8 shows that a large amount of data is associated with a good SNR even if the ice crystals are not large enough to produce non-Rayleigh scattering.

- Figure 4e: Why is this plot dark?
  (ii) As was written in the text and the caption of Fig. 4, the shading allowed to highlight the Rayleigh reflectivity region at cloud top (non-shaded region) which is used to derive the two-way differential path-integrated attenuation.
  (iii) To make it more readable, we change this panel (Now Fig. 4c) and now, the shading actually indicates the Rayleigh reflectivity region.
- Line 213-214: I cannot identify vertical stripes. Can you add marks to the figure?
  (ii) It is not realistically possible to add marks to the figure because the vertical bands are present over almost the whole panels 4b and 4d. (iii) Instead, as suggested by the reviewer 1, we replaced the text to be more explicit by "vertical bands that alternate between more or less dark blue/red.
- Figure 9: How did you estimate the standard deviation from one profile?
   (ii) Standard deviation was probably not the right wording, we meant

uncertainties. The derivation of uncertainties is fully described in the retrieval methodology of section 4.5. Retrieving the uncertainties is the object of the 3rd point of the methodology: it is done via Monte Carlo propagation for parameters mu and Dm, and the errors of the other parameters are obtained via error propagation as described in the 4th and 5th points. (iii) In the caption of Figure 9, we replaced standard deviation by uncertainties and recalled how they are derived.

---

## Author Response (AR2)

**acp-2022-136**

**Responses to 2nd review**

Highly supercooled riming and unusual triple-frequency radar signatures over McMurdo station, Antarctica

F. Tridon, I. Silber, A. Battaglia, S. Kneifel, A. Fridlind, P. Kalogeras, and R. Dhillon

August 3rd, 2022

We thank the reviewer 2 for his renewed efforts and time for providing additional technical comments which greatly help polishing the manuscript. All our point-to-point answers are highlighted in red below according to the following sequence: (i) comments from referees/public, (ii) author's response, and (iii) author's changes in manuscript.

**General comment:**

I am satisfied with the authors responses. The added descriptions, especially about the environments around the sites, are helpful for me. I have a few minor suggestions and questions.

**Specific comments:**

1. (i) It would be worth mentioning about minimum detectable reflectivity for the corrected data for all radars. This may be a good reference for future studies.
(iii) We added the sensitivity of the radars after calibration in the 3rd paragraph of section 2.2.

2. (i) Figure 1: Data sample size at each temperature also helps.
(iii) We added panels to show the number of samples per temperature level.

3. (i) Section 4.3 "130000 observations": Does "observations" mean range-gate data points or profiles?
(ii) We meant range gate data. (iii) We replaced "observations" by "data points (with resolution of 2 s by 30 m)" in the manuscript.

4. (i) I might be confused with the definition of diameter D in this paper. Does "D" or diameter refer to water equivalent diameter or maximum dimension of irregular shapes, or others?
(ii) In this study, D refers to the maximum dimension of ice particles. (iii) We added this information just after equation (1) and in the caption of figure 12.

5. (i) I think that I still do not understand why radar misalignment is emphasized at lower levels (smaller sampling volumes).

(ii) Indeed, at lower levels the sampling volumes mismatch is mainly due to the slightly different location of the radars: since the beam widths are small and the radars are nearly 10 meters apart, there is less overlap between the sampling volumes of the different radars. When the cloud is highly heterogenous, this can explain the unusual DWRs below 1 km.